# The Labyrinth of Links: Navigating the Associative Maze of Multi-modal LLMs

**Hong Li, Nanxi Li, Yuanjie Chen, Jianbin Zhu, Qinlu Guo, Cewu Lu, Yong-Lu Li**[*]
Shanghai Jiao Tong University, Shanghai Innovation Institute
{hong_li,andyc_03,cyj2003,bin_pig,guoqinlu,lucewu,yonglu_li}@sjtu.edu.cn

## Abstract

Multi-modal Large Language Models (MLLMs) have exhibited impressive capability. However, recently many deficiencies of MLLMs have been found compared to human intelligence, *e.g.*, hallucination. To drive the MLLMs study, the community dedicated efforts to building larger benchmarks with complex tasks. In this paper, we propose benchmarking an essential but usually overlooked intelligence: **association**, a human's basic capability to link observation and prior practice memory. To comprehensively investigate MLLM's association performance, we formulate the association task and devise a standard benchmark based on adjective and verb semantic concepts. Instead of costly data annotation and curation, we propose a convenient **annotation-free** construction method transforming the general dataset for our association tasks. Simultaneously, we devise a rigorous data refinement process to eliminate confusion in the raw dataset. Building on this database, we establish three levels of association tasks: single-step, synchronous, and asynchronous associations. Moreover, we conduct a comprehensive investigation into the MLLMs' zero-shot association capabilities, addressing multiple dimensions, including three distinct memory strategies, both open-source and closed-source MLLMs, cutting-edge Mixture-of-Experts (MoE) models, and the involvement of human experts. Our systematic investigation shows that current open-source MLLMs consistently exhibit poor capability in our association tasks, even the currently state-of-the-art GPT-4V(vision) also has a significant gap compared to humans. We believe our benchmark would pave the way for future MLLM studies. *Our data and code are available at:* https://mvig-rhos.com/llm_inception.

## 1 Introduction

Multi-modal Large Language Models (MLLMs) have recently made significant breakthroughs in perceiving diverse modality input and solving a broad range of tasks Zhang et al. (2024a); Carolan et al. (2024). As GPT-4V(ision) Achiam et al. (2023) and Gemini Team et al. (2023); Reid et al. (2024) address challenges that researchers have been exploring for a considerable period. Subsequently, numerous researchers have developed diverse open-source MLLMs AI et al. (2024); Bai et al. (2023b); Wang et al. (2024b); Dong et al. (2024); Liu et al. (2023a); Li et al. (2024a); Ye et al. (2023; 2024). These models usually use the Large Language Model (LLM) as the core component and expand to multi-modal with a specific module Yin et al. (2023) that transfers multi-modal tokens into language tokens, achieving alignment between different modality encoders.

MLLMs demonstrated ability in visual reasoning, which requires understanding the input query and then making judgments based on the visual content. Much prior work has been dedicated to evaluating the levels of their visual reasoning capabilities. However, to the best of our knowledge, how to evaluate the association ability of MLLMs is overlooked. **Association** is one of the most fundamental capabilities of human intelligence. It provides the foundation for creative thinking and helps humans to summarise scattered information into structured knowledge to enhance memory and understanding Mednick (1962); Ausubel (1963), perception, rule discovery, embodied AI, *etc*.

---

[*]Correspondence to: Yong-Lu Li <yonglu_li@sjtu.edu.cn>.

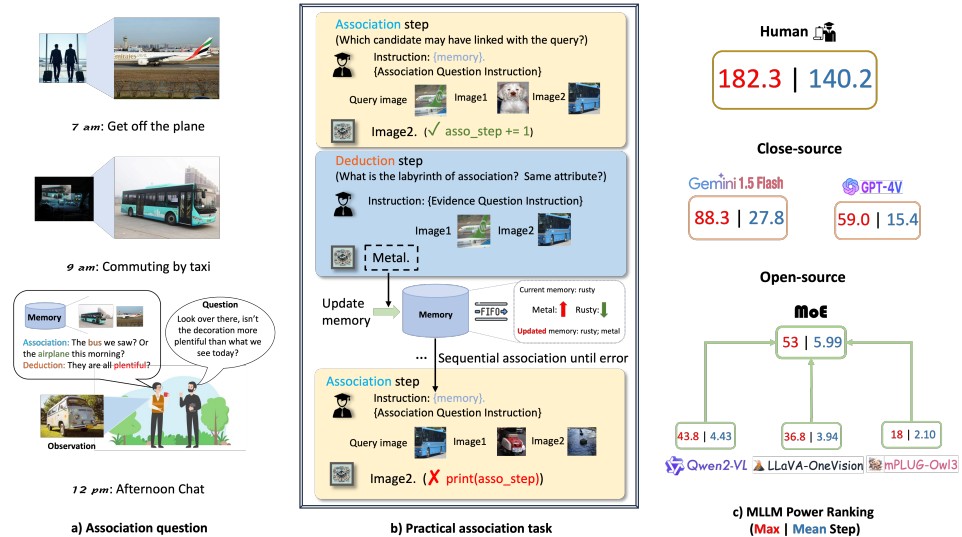

Figure 1: Our insight and proposed association task. a) Association is always in our lives. b) Our proposed practical association task. c) The performance of current MLLMs and human experts. The results demonstrate a significant gap between current MLLMs and humans in association tasks.

In this paper, we aim to devise a standard benchmark to evaluate MLLM's capability in association tasks. As shown in Figure 1a, we always connect observations with previous practice memory. This association is usually the semantic concept shared between different objects, representing common features in general dataset samples. For instance, the "painted airplane" and "painted bus" share the same attribute of "painted", which can be seen as an association chain in the association task. Next, it is natural to ask *can we design a general method to build an association benchmark based on existing datasets without too much extra effort?* To this end, we propose an annotation-free association construction method, which easily transfers the general dataset for the association task. As in Figure 1b, we devise a standard association benchmark using the object's concepts as an association chain. Given the benchmark, we comprehensively investigate MLLMs' capabilities in association tasks across multiple memory strategies, multiple open-source and closed-source MLLMs, as well as cutting-edge Mixture-of-Experts (MoE) and human expert testers.

Extensive experiments demonstrate that current open-source MLLMs consistently have a weak ability in association tasks, and even the closed-source GPT4-V Achiam et al. (2023) and Gemini-1.5-Flash Reid et al. (2024) also have a huge gap from human performance. As Figure 1c shows, the best closed-source Gemini-1.5-Flash attains an average mean-step of $27.8$, while humans attain $140.2$ in the attribute of adjective concepts. As a complementary of existing benchmarks, we believe our benchmark and baselines will pave the way for human-like intelligent agents.

In conclusion, our main contributions are: **1)** To evaluate the association ability of MLLMs, we propose a new multi-modal task and a corresponding convenient annotation-free construction method that can transform the general dataset for association tasks within various semantic concepts. **2)** We devise a standard association benchmark based on adjective semantic concepts within objects and verb semantic concepts within actions and further evaluate MLLMs' performance through extensive experiments. **3)** We systematically investigate MLLMs' capability on association tasks through tuning-free methods and propose potential future directions for association tasks.

## 2 RELATED WORKS

**Multi-Modal Visual Reasoning.** Various works are dedicated to understanding and reasoning about the semantic concept between multi-images. Several works, such as Visual Genome Krishna et al. (2017) and Bongard-HOI Jiang et al. (2022), target human-object interaction (HOI) tasks to investigate the visual relationship between different objects. Recent work Zhang et al. (2024b) investigates the MLLM's ability in low-level perception question-answering and description tasks with paired input images. Despite their success in perceiving and understanding multi-images, these methods are confined to single-step evaluation, lacking the investigation of multi-step association ability.

**Multi-Modal Large Language Model.** There has been a surge of MLLMs in the deep learning community, which use off-the-shell LLMs to support multi-modal inputs and demonstrate promise

for zero-shot generalization. For instance, LLaVA Liu et al. (2023a) InstructBLIP Dai et al. (2023), InternLM-XComposer Dong et al. (2024), Qwen-VL Bai et al. (2023a), MiniGPT-4 Zhu et al. (2023), and mPLUG-Owl2 Ye et al. (2023) make leading in last year according to their powerful visual perception capabilities. More recently, these models Wang et al. (2024b); Li et al. (2024a); Ye et al. (2024) are updated to improve their ability in multi-images or video input. In this work, we mainly evaluated the leading open-source models Qwen2-VL Wang et al. (2024b), mPLUG-Owl3 Ye et al. (2024), and LLaVA-OneVision Li et al. (2024a).

**Tuning-Free Engineering.** The development of MLLMs has led to a significant improvement in in-context learning Doveh et al. (2024); Wies et al. (2024). Additionally, the use of visual and language prompts has been shown to enhance the performance of MLLMs. Research has indicated that visual cues such as listed numbers Yan et al. (2024) and shapes Shtedritski et al. (2023); Mani et al. (2020) can be effectively understood by MLLMs. Similarly, carefully designed language prompts Liu et al. (2023b), Chain-of-Thought (CoT) Wei et al. (2024), tree-of-mixed-thought (ToMT) Hu et al. (2023), have shown promising results. Common knowledge Shao et al. (2023); Liu et al. (2024a) can be seen as an instruction of Prior knowledge about how to make decisions on a task. As a result, we chose to employ the general prompt-engineer one-shot, CoT, and Common knowledge.

## 3 ANNOTATION-FREE ASSOCIATION CONSTRUCTION

For a general dataset with $N$ samples, it can be formed as $\{(x_i, y_i) \mid x_i \in X, y_i \in \{c_1, \ldots, c_n\}, i = 1, 2, \ldots, N\}$. Let $x_i = (x_i^{m_1}, \ldots, x_i^{m_k})$ be a sample with $k$ modalities, more specifically, for a task with individual modality, the sample is described as $x_i = (x_i^{m_1}, )$. $y_i$ is the annotation for sample $x_i$, which may existed in various granularities. These annotations indicate a semantic concept, with object categories as nouns, actions as verbs, as well as attributes, and affordance as adjectives.

Given this definition, labels represent a subset of link concepts present in the given sample. The core of human association, on the other hand, involves identifying the overlapping concepts between newly acquired observations and prior practice memory. Hence, it is intuitive to use annotations from the raw dataset to construct the association chain, which reflects the common concepts shared by two samples, such as the presence of the *"shoot ball"* action in both samples. In the following, we first generate possible chains for the association task, then create the ground-truth labyrinth to reason the chain behind the association.

**Generating Semantic Concept Association Chain.** Association connects objects by any potential links. We defined associations existing when any pre-defined semantic concept appears between objects. To this end, for a general dataset $\{(x_i, y_i) \mid x_i \in X, y_i \in \{c_1, \ldots, c_n\}, i = 1, 2, \ldots, N\}$, we randomly select two samples to form a new sample pair for association. If the selected samples have *identical labels or share at least one common label*, we assign a corresponding label of 1; otherwise, we assign a label of 0. Hence, we get the new association dataset as:

$$\{(x_i, x_j, z_{ij}) \mid x_i, x_j \in X, 1 \le i < j \le N\}, \text{ where } z_{ij} = \begin{cases} 1 & \text{if } y_i \cap y_j \neq \emptyset \\ 0 & \text{otherwise} \end{cases}. \qquad (1)$$

In this way, for each sample $x_i$ in the original dataset, we construct a positive association set with $K$ positive samples, $\boldsymbol{x}_i^+ = [p_i^1, \ldots, p_i^K]$, where at least one potential association concept exists. Additionally, we devise another negative association set with $L$ samples, $\boldsymbol{x}_i^- = [q_i^1, \ldots, q_i^L]$, in which no predefined association chain present.

**Deducting the Evidence of Association Chain.** Associations use implicit links to connect objects, which make correct decisions with unspecified links. To uncover the labyrinth, we devise deduction steps that reason the association links after association. This evidence will fused into memory and influence subsequent steps. To realize this, we collected the full set of shared concepts within all possible positive set $\boldsymbol{x}_i^+$ as $\hat{C} = \bigcup_{i=1}^N \{z_{ij} \mid z_{ij} = y_i \cap y_j, 1 \le i < j \le N\}$. In this setting, the dataset deducting the evidence of the association chain is depicted as:

$$\{(x_i, p_i^k, \boldsymbol{s}_i), \mid x_i, p_i^k \in X, p_i^k \in \boldsymbol{x}_i^+, \boldsymbol{s}_i = \{s_i^1, \ldots, s_i^R\} \subset \hat{C}\}, \qquad (2)$$

where $p_i^k$ represents the correctly predicted positive sample at the stage of chain association, and $\boldsymbol{s}_i = \{s_i^1, \ldots, s_i^R\}$ denote the $R$ common concepts.

**Evaluation of the Association Task.** Based on the above definition, we can easily construct the association task. First, we randomly select one sample, $x_{\text{query}}$, as the initial query image. Then, for each step $t$, We randomly choose candidate samples $p_i^t$ and $q_i^t$ from the positive $\boldsymbol{x}_{\text{query}}^+$ and negative $\boldsymbol{x}_{\text{query}}^-$ sets, respectively. Next, we deliver new observations and previous practice memory into MLLMs determining which candidate shares common concepts with the query, the output of $t$-th step described as $o_t$. If the model makes the correct decision, $o_t = p_i^t$, the query and correctly predicted images will be used to deduct the chain of association that updates memory. Then convert to the next association step, the correctly predicted samples taken as the query image and repeatedly select candidate samples to conduct association and deduction progress. If make an error, we calculate the maximum association steps and exit. The above description can be depicted as:

$$t = \text{step} \begin{cases} 1 + \text{step}(\mathcal{F}(x_{\text{query}}, (\boldsymbol{x}_{\text{query}}^+, \boldsymbol{x}_{\text{query}}^-))) \text{ and } x_{\text{query}} = p_i^t & \text{if } o_t = p_i^t, \\ t, & \text{otherwise (output } t \text{ and exit)}, \end{cases} \quad (3)$$

where $x_{\text{query}} \in \{x_1, \dots, x_N\}$ and $\mathcal{F}$ represents the forward computation of MLLMs. step is the computation of combined steps of association and deduction.

# 4 ASSOCIATION BENCHMARK

In this section, we introduce a benchmark based on the semantic concepts of adjectives and verbs, *i.e.*, object attributes and affordances, and human actions. Specifically, we utilize the annotation-free construction method proposed in Section 3 on the Object Concept Learning (OCL) Li et al. (2023) to generate an attribute and affordance association datasets, and on the Pangea Li et al. (2024c) to generate action association dataset [1]. In the following, we comprehensively investigate the association ability from the single-step association, synchronous association, and asynchronous association settings as shown in Figure 2.

## 4.1 CONSTRUCTING ASSOCIATION TASK

### 4.1.1 SINGLE-STEP ASSOCIATION

The association refers to the link between the current observation and prior practice memory (Figure 2a). A single-step association represents one phase within a broader association task, where memory is indispensable in decision-making. In this case, we include the correct memory to simulate prior practice, guiding the decision-making process. In the experiment, we compute the association and deduction success ratio as the main single-step association metric (Section 5.1.2).

### 4.1.2 SYNCHRONOUS ASSOCIATION

Synchronous association, where each step adheres to the same principle throughout the entire process, is a core capability of human intelligence (Figure 2b). With this ability, humans can gradually unveil the underlying rules of the task and reduce the likelihood of errors. Utilizing our constructed association dataset, we take different input-paired samples with the same association concepts $\hat{c}_t$ to evaluate the synchronous association. In this setting, the association dataset can be depicted as:

$$\{(x_i, x_j, z_{ij}) \mid x_i, x_j \in X, 1 \le i < j \le N\}, \text{ where } z_{ij} = \begin{cases} 1 & \text{if } \hat{c}_t \subset \{y_i \cap y_j\} \\ 0 & \text{if } y_i \cap y_j = \emptyset \end{cases}. \quad (4)$$

It is worth noting that current MLLMs lack memory during the inference stage, relying solely on the input samples. To address this problem, we introduce a memory base to imitate the human's memory in the synchronous association. Specifically, we transfer the inference process into the memory base after each step. The updated memory and the input sample are then used for the next step. In the experiment, we compute the Max | Mean steps metric to evaluate the model (Section 5.1.2).

### 4.1.3 ASYNCHRONOUS ASSOCIATION

When there are multiple principles in games, the underlying principle will gradually change as the game progresses. For example, the first two steps with "metal" as the chain, then "furry" and again

---

[1]To further demonstrate the capability on verb concept, we further implemented on action HMDB Kuehne et al. (2011) datasets. The results are included in the section F of the supplementary material.

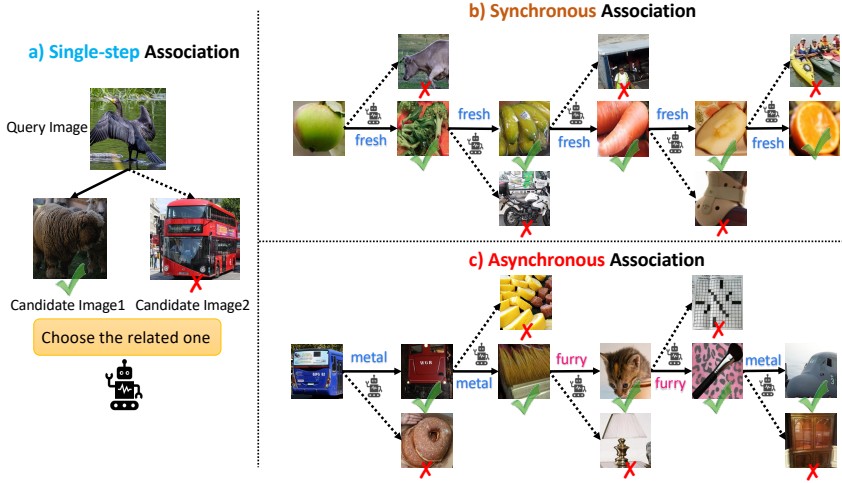

Figure 2: Semantic figure of association question. (Left) Single-step association with a fixed correct memory. (Right) Synchronous and asynchronous association with dynamic memory. Synchronous involves one category in the chain, while asynchronous improves complexity with two categories.

"metal", constituting a loop strategy (Figure 2c). In this case, we need to call back the memory from the previous step instead of the last step. Similar to the synchronous association, we utilize the association dataset for the asynchronous association setting by selecting multiple shared concepts between different input samples. We formulate the dataset as:

$$\{(x_i, x_j, z_{ij}) \mid x_i, x_j \in X, i < j, i, j = 1, 2, \ldots, N\}, \text{ where } z_{ij} = \begin{cases} 1 & \text{if } \hat{C}^m \cap \{y_i \cap y_j\} \\ 0 & \text{if } y_i \cap y_j = \emptyset \end{cases}, \quad (5)$$

where $\hat{C}^m$ indicates $m$ shared concepts selected from $\hat{C}$. In addition, we also utilize the memory base that the strategy is the same as the synchronous association for the asynchronous association.

### 4.2 DATA REFINEMENT FOR ASSOCIATION TASK

Depending on annotation-free association construction methods, we transfer the general dataset for the association task. The dataset has paired input samples and is labeled with whether they have common concepts for the association step. Furthermore, paired input samples with shared concepts were created for the deduction step. While these support the challenges in association tasks, there is still a possibility of confusing samples. To address this, we introduce a data refinement method.

We implemented a three-step strategy to ensure data quality, including an Image resolution filter, MLLM verification, and Human expert evaluation. Specifically, the Image resolution filter screened out all images with less than $50,000$ pixels to ensure superior visual quality. The MLLM verification [2] takes a question-answer strategy with OpenAI's GPT4-V Achiam et al. (2023) and Google's Gemini-1.5-Flash Reid et al. (2024) to ensure each annotation of raw data exists in the image. Then, the human expert evaluation is conducted through our custom-designed interface, enabling testers to complete the association task and eliminate low-quality samples or those with ethical concerns [3].

Our benchmarks are implemented in adjective concepts and verb concepts, which include attribute and affordance in OCL Li et al. (2023) and action in Pangea Li et al. (2024c). In the OCL dataset, we selected eight attributes with good perception performance, such as "metal, ripe, fresh, natural, cooked, painted, rusty, furry", and eight affordances "sit, imprint, push, carry, cut, clean, open, break". In addition, we selected eight actions of "run, hit, drive, dress, cooking, build, shake, cut".

### 4.3 BASELINE FOR ASSOCIATION

There are various methods to improve the concept perception. Our focus lies in exploring tuning-free methods, which harness the inherent capabilities of the model. To this end, we employ popular prompt engineers to improve the understanding of MLLMs, including common knowledge (Com-Know), one-shot, and chain-of-thought (CoT).

---

[2]We compare the performance with and without the MLLM verification in the section G of supplementary.
[3]The detailed visualization and description refer to Subsection A.1 in the supplementary.

For the memory base in association, we introduce three different memory strategies that transfer each inference process into memory after each association step, including structure memory (StructM), natural language memory (NLM), and chain memory (ChainM). StructM and NLM both simulate human memory, differing only in their descriptive approaches. They incorporate memory attention mechanisms to determine whether to reinforce or forget memories. Specifically, The underlying strategy of memory base is that if one type of memory knowledge $m_k$ appears in evidence at one step, we add a repetition weight $w_r$ to the attention weight W; otherwise, the attention weight decays, *i.e.*, subtracts forgetting decrement $d_f$. This strategy is described as:

$$\mathrm{W}_{m_k} = \begin{cases} \mathrm{W}_{m_k} + w_r, & \text{if } m_k \in \text{evidence} \\ \mathrm{W}_{m_k} - d_f, & \text{otherwise} \end{cases}. \tag{6}$$

StructM uses raw structure memory as input, while NLM transforms it into descriptions that are more aligned with human language. Additionally, ChainM is more closely aligned with the task, as it stores the previous inference process as a chain. In the ChainM strategy, each inference process is treated as a subchain, represented as $obj1 -> concept -> obj2$, and is concatenated into the prior memory. The detailed description and practice examples refer to section A.2 in the supplementary.

## 5 EXPERIMENTS

### 5.1 SETTINGS

#### 5.1.1 IMPLEMENTATION DETAILS

We systematically conduct experiments that include concept perception, single-step, synchronous, and asynchronous association. Specifically, we implement the task as a multi-choice setting in all experiments, taking one query image and two candidate images with one correct as input and output correct image index. Based on this, we first devise preliminary concept perceptions that involve popular prompt engineering skills on open-source MLLMs [4] to investigate perception capabilities in attribute concept. Then, we convert to evaluate MLLM's association capabilities that make decisions based on current observation and prior practice memory, *i.e.*, input with additional content of previous practice. We develop single, synchronous, and asynchronous associations according to fixed or dynamic memory. The single-step association means the model decides with observation and correct prior practice. Meanwhile, the synchronous and asynchronous association set model at a dynamic task that iteratively arrives at a decision and then deducts the underlying evidence, which means the memory may have wrong information for the next judge.

For single-step, synchronous, and asynchronous association, we design three types of memory bases, *i.e.*, Structure Memory (StructM), Natural Language Memory (NLM), and Chain Memory (ChainM). In addition, we involve the baseline of No Memory (NoM) which means determining at a dynamic setting without the memory base. For the detailed description of the type of memory base and the usage in the prompt, please refer to the section A.2 in the supplementary.

We utilize three open-source MLLMs in preliminary concept perception: QWen-VL Bai et al. (2023b), LLaVA-NeXT-7B Liu et al. (2024a), and LLaVA-NeXT-13B. For formal association, we utilize three new-versions MLLMs that break through the MLLM's capabilities in multi-images: LLaVA-OneVision Li et al. (2024a), QWen2-VL Wang et al. (2024b), and mPLUG-Owl3 Ye et al. (2024). Besides, we evaluate the performance of Mixture-of-Experts (MoE) that combined three open-source MLLMs. In experiments, open-source MLLM is run on a single NVIDIA A100 80G GPU. Apart from open-source MLLMs, we include the evaluation of the closed-source MLLMs of GPT4-V Achiam et al. (2023) and Gemini-1.5-Flash Reid et al. (2024). Simultaneously, we involve the results of three human experts to demonstrate the gap between MLLM with human intelligence.

#### 5.1.2 METRICS FOR ASSOCIATION TASK

**Max | Mean Step.** In an association task, *max-step* indicates the maximum number of steps in one round of association, *i.e.*, the maximum length of a correctly predicted association chain. While *mean-step* refers to the average maximum association step across multi-rounds of association tests.

---

[4] For the detailed description of the setting of concept perception, please refer to section D.

| Type | Concept | Memory | Models (Success Ratio) | | | | | |
|---|---|---|---|---|---|---|---|---|
| | | | LLaVa-OneVision | QWen2-VL | mPLUG-Owl3 | **Gemini-1.5-Flash** | **GPT-4o** | Avg. |
| Association Success Ratio | Attribute (Adjective) | NoM | 75.52 | 76.36 | 54.05 | 75.40 | 84.49 | 73.16 |
| | | StructM | 78.41 | 77.74 | 72.67 | 87.30 | 88.83 | 80.39 |
| | | NLM | 80.40 | 86.63 | 73.33 | 88.39 | **89.87** | 83.72 |
| | | ChainM | 77.73 | 73.55 | 70.70 | 83.78 | 78.86 | 76.92 |
| | Affordance (Adjective) | NoM | 70.61 | 75.10 | 53.80 | 77.78 | 84.93 | 72.44 |
| | | StructM | 73.39 | 75.39 | 73.02 | 84.45 | 85.36 | 61.25 |
| | | NLM | 76.13 | 79.37 | 67.79 | 85.40 | **86.76** | 79.10 |
| | | ChainM | 74.38 | 82.67 | 68.46 | 80.02 | 81.54 | 77.40 |
| | Action (Verb) | NoM | 75.74 | 78.43 | 57.94 | 84.21 | 86.97 | 76.66 |
| | | StructM | 75.44 | 82.10 | 73.92 | 88.10 | 88.72 | 81.66 |
| | | NLM | 78.66 | 88.01 | 70.04 | **89.58** | 86.13 | 82.48 |
| | | ChainM | 76.92 | 85.59 | 69.50 | 87.58 | 85.90 | 81.10 |
| Deduction Success Ratio | Attribute (Adjective) | | 49.38 | 58.11 | 61.21 | 65.82 | 78.14 | 62.53 |
| | Affordance (Adjective) | | 21.07 | 15.98 | 33.30 | 33.08 | 27.30 | 26.15 |
| | Action (Verb) | | 46.61 | 49.99 | 42.89 | 55.69 | 57.28 | 50.60 |

Table 1: Mean success ratio of each concept on single-step association with four memory strategies across open-source and close-source MLLM. The best and second results of each concept are shown in **bold** and underline, respectively. For detailed results on each category refer to Table 5, 6, 7.

**Success Ratio.** For concept perception and single-step association, suppose the total number of samples of each concept $\hat{c}_i$ is $T_i$, and the correctly judged based on the shared concept is $T_i^+$ samples. For any association and deduction step, the success ratio on concept $\hat{c}_i$ is defined as $r_i^+ = \frac{T_i^+}{T_i}$.

## 5.2 RELATIONSHIP BETWEEN PERCEPTION AND ASSOCIATION

In this subsection, we investigate the MLLM's ability in attribute concept perception that is implemented on the OCL Li et al. (2023). We devise multiple task complexity, including individual perception and combination perception. The detailed description and results are shown in section D in the supplementary. These results indicate that while popular prompt engineering skills improve performance in specific cases, *i.e.*, prompt engineering skills compared to Normal, they still exhibit limited capabilities in the widely-used object understanding benchmark. For instance, in the individual perception, the highest perception success ratio of $0.690$ reflects an improvement of $0.190$ over the random baseline of $0.5$. Furthermore, the combination perception attains the success ratio of $0.542$, improving $0.292$ over the random baseline of $0.25$.

The association is a capability built upon foundational perception abilities. Hence, this prompts us to reduce the complexity of perception and *disentangle* the evaluation of perception from association. In practice, we selected categories with good perception performance for association tasks. This enables a more rigorous assessment of the MLLM's associative capabilities.

## 5.3 RESULT OF ASSOCIATION

Based on the finding of concept perception, we convert to evaluate MLLM's association capability. In the following subsection, we first access MLLM's ability in the single-step association that makes decisions with observation and correctly fixed memory. Then, we evaluate its associative capabilities in a dynamic setting, with evidence derived from the MLLM's deduction. We further investigate the efficacy of MLLMs in a straightforward synchronous association encompassing only one semantic concept within the association chain. Furthermore, we increase the task complexity to attain asynchronous association by involving *two* semantic concepts in the association chain.

### 5.3.1 SINGLE-STEP ASSOCIATION

Table 1 shows the mean result in adjective and verb concepts, *i.e.*, attribute and affordance on the OCL Li et al. (2023) and verb on the Pangea Li et al. (2024c). We focus on the metric of "success ratio" in this part, and the detailed result on each item can be found in supplementary Table 5, 6, 7.

The result shows that GPT4-o achieves the highest performance in adjectives and Gemini-1.5-Flash makes advances in verbs, but those all remain a certain gap from humans, *i.e.*, success ratio compared to $1$. It deserves noted that humans are unlikely errors as association links are given in context. Moreover, our comprehensive data refinement ensures the data quality that prevents any errors induced by annotations confusion. Besides, the association has certain improvements across various concepts, *i.e.*, the proposed memory strategy is higher than NoM. Simultaneously, the NLM at-

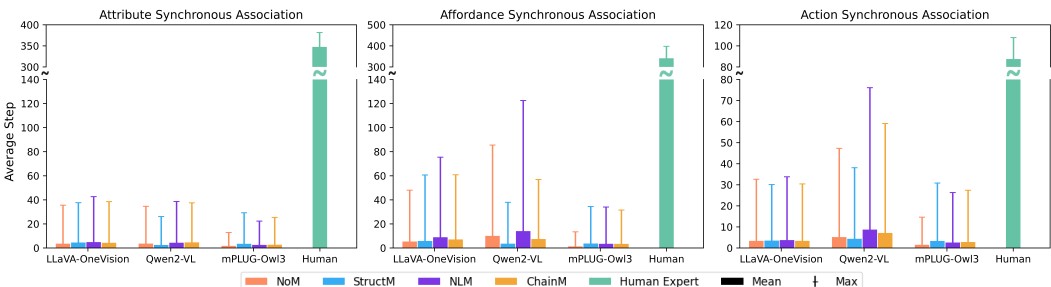

Figure 3: The average Max | Mean step on the individual concept synchronous association across three open-source MLLM and humans. Different columns within each MLLM indicate different memory strategies. For detailed results on each category refer to Table 8, 9, 10 in the supplementary.

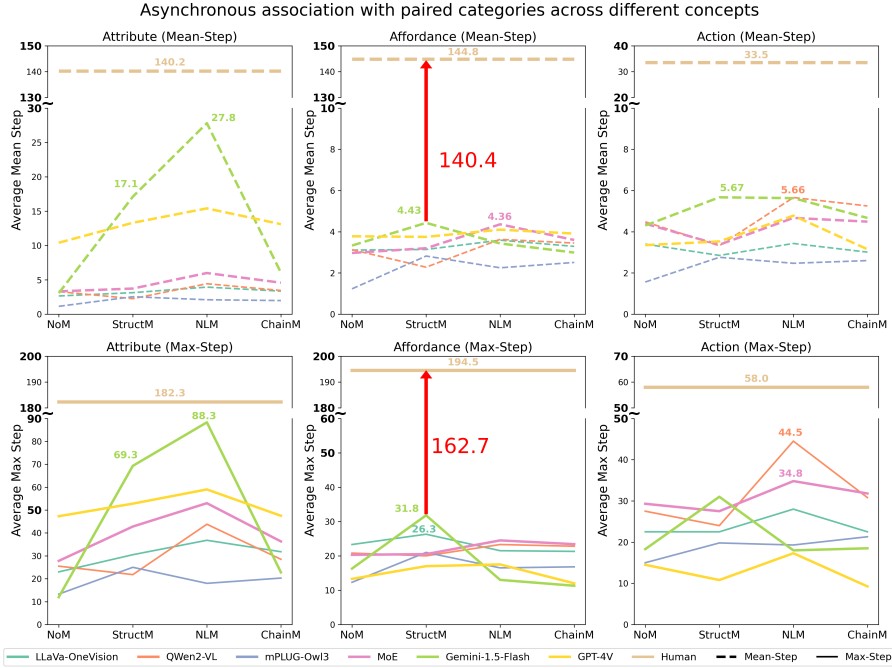

Figure 4: The average Max | Mean step of asynchronous association with paired categories across different concepts. The upper subfigure is mean-step, while the lower subfigure is max-step. The detailed results on each different category group refer to Table 11, 12, 13 in the supplementary.

tained the strongest performance, we speculate that due to the current MLLM being trained on a large amount of internet visual language data, lacking structured data training.

### 5.3.2 SYNCHRONOUS ASSOCIATION

The primary focus of our design revolves around the stability of MLLMs in synchronous association within multi-step settings. The Max | Mean step metrics serve as direct measures of the length of a continuous association chain, offering more precise insights into the capacity to comprehend rules and uphold stability. Intuitively, when humans master the rules through some examples or previous association processes, they can keep going without making mistakes.

Figure 3 shows the attribute, affordance, and action results in the synchronous association with four memory strategies. The results clearly show a significant gap between open-source MLLMs and human experts, *i.e.*, the green column compared to others. In attribute synchronous association, human experts achieve the mean-step of 350, while the average step of the open-source models is below 20. Additionally, different memory strategies exhibit a consistent trend with single-step associations across various approaches, with NLM showing slightly better performance in both max and mean step evaluations compared to other memory strategies. Furthermore, the open-source QWen2-VL outperforms LLaVA-OneVision and mPLUG-Owl3, which we attribute to its pre-training methods that achieve stronger cross-modal alignment, resulting in greater performance gains. This guides our

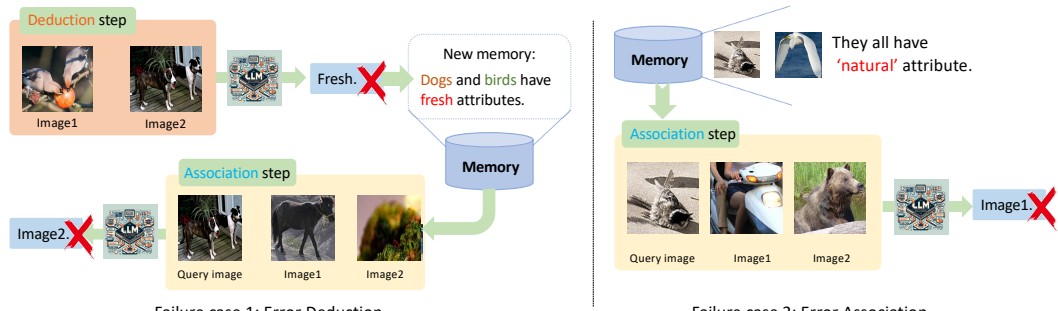

Figure 5: Failure cases of GPT4-V in asynchronous association with paired attribute categories. (Left) The error arises from the deduction. (Right) The error originates from the association.

future exploration toward improving the associative capabilities of MLLMs. The detailed analysis of comprehensive results refers to supplementary.

### 5.3.3 Asynchronous Association

In the following, we proceed to investigate the MLLM's ability in asynchronous association. That improves the complexity by introducing two different semantic categories compared to synchronous association. Figure 4 summarises the results of asynchronous association in the OCL and Pangea datasets. We involve four different semantic concept pairs for each dataset. In all datasets, we take the same memory strategies as the synchronous association.

From the result, we can easily found asynchronous association can be a challenge for humans in some cases. For instance, in the Pangea action, the human expert achieves an average mean-step of $33.5$ and an average max-step of $58.0$, which is lower than synchronous association. In addition, it meets our expectation that MoE outperforms the individual open-source MLLM in all cases, *i.e.*, the max-step and mean-step of MoE are higher than open-source MLLMs. It catches our attention that closed-source MLLM has a close performance to open-source MLLM, *i.e.*, GPT4-V, and Gemini-1.5-Flash compared to open-source MLLMs. This indicates that all current models exhibit poor performance on association tasks when the gap in perceptual capabilities is minimized. Furthermore, Gemini-1.5-Flash outperforms GPT-4V in certain cases, which we speculate is due to the MLLM verification step in data refinement. This step employs a question-answer strategy to verify the existence of concepts, initially relying on Gemini-1.5-Flash for judgment and deferring to GPT-4V only when Gemini-1.5-Flash is unable to provide a decision (Subsection 4.2).

### 5.4 Analysis of Results

In the following, we first discuss some failure cases and then deeply analyze the potential reasons why MLLMs are inferior to humans via attention and underlying capabilities [5].

**Analysis of Failure Case.** We divided the errors into two types according to the stage in our benchmark, *i.e.*, Error deduction, and Error Association. For error deduction, the MLLM predicts the error links for associated objects. This then causes the error in the association step (Left of Figure 5). Conversely, for error association, MLLM has the correct memory and makes errors due to the limited perception capability (Right of Figure 5). More interestingly, these failures are consistent with humans, we may derive error information and further induce incorrect judgments.

**Analysis of Attention.** We demonstrate that there is a significant gap between MLLMs and humans. We speculate their two main factors: a lack of multi-image instruction tuning and limitations in contextual understanding. The first statement is also observed from existing work Song et al. (2024); Wang et al. (2024c), that the current MLLM has a weak ability in multi-image understanding since the lack of its instruction data. In addition, recent research Wang et al. (2024a) has demonstrated MLLMs' poor performance in handling long contexts, *i.e.*, memory. We also visualize the attention map of open-source MLLM Qwen-VL on OCL attribute concept to support this observation, as Figure 6, we can easily find that response has predominate attention at the position close to response instead of the part for decision-making in both StructM and NLM.

---

[5]Due to limited space, the detailed analysis refers to Section C in the supplementary.

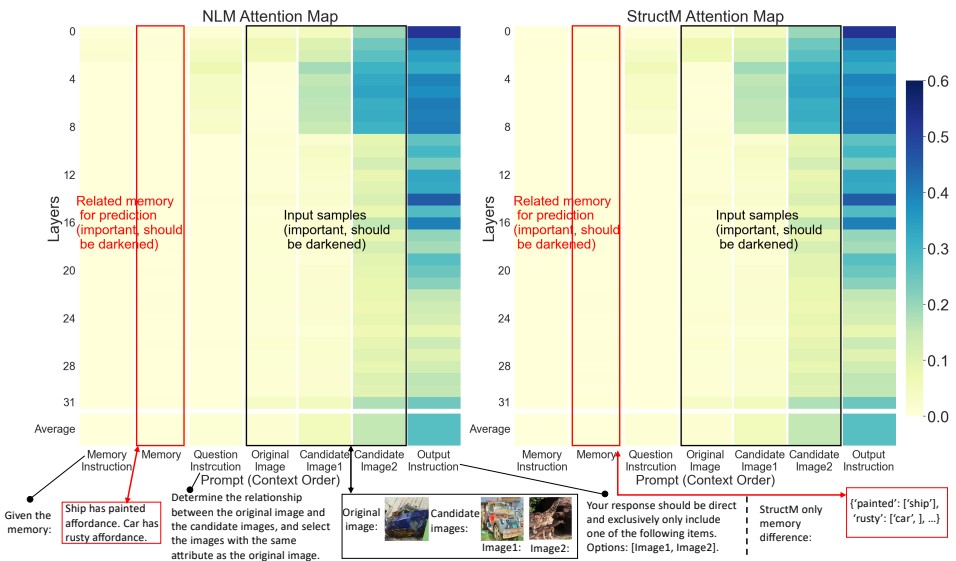

Figure 6: Attention weight of QWen-VL with StructM and NLM strategies in asynchronous association. A deeper color means greater attention. The frame marks the critical area in the input context.

## 5.5 ABLATION STUDY

We conduct ablation studies that take different example sizes in the associations to analyze the selection of an initialized memory base. Table 2 summarises the results for example sizes of 1, 3, and 5 in the LLaVA-OneVision Li et al. (2024a) and QWen2-VL Wang et al. (2024b). Regardless of whether considering the max or mean step, the influence of different sample sizes is minimal, with an average gap of 1.51 for the maximum step and 0.05 for the mean step. As the maximum step indicates peak performance in some cases, while the mean step reflects more stable performance, we have chosen a sample size of 3 based on the mean step results for all our experiments.

| Example size | LLaVa-OneVision | | | QWen2-VL | | | Avg. | |
|---|---|---|---|---|---|---|---|---|
| | StructM | NLM | ChainM | StructM | NLM | ChainM | | |
| 1 | 30.3 \| 3.22 | 33.3 \| 3.92 | 28.3 \| 3.34 | 21.5 \| 2.05 | 36.8 \| 4.36 | 27.8 \| 3.41 | 29.67 \| 3.38 | |
| 3 | 26.3 \| 3.14 | 36.8 \| 3.94 | 31.8 \| 3.35 | 21.8 \| 2.26 | 43.8 \| 4.43 | 22.8 \| 3.45 | 30.55 \| **3.43** | |
| 5 | 32.5 \| 3.17 | 36.5 \| 3.90 | 32.8 \| 3.35 | 21.5 \| 2.09 | 38.8 \| 4.53 | 25.0 \| 3.39 | **31.18** \| 3.40 | |

Table 2: The ablation of example sizes in the association. (Detailed results in Table 14.)

## 6 LIMITATIONS AND DISCUSSION

In this paper, our investigation is limited to MLLMs' zero-shot ability in association tasks across adjectives and verb semantic concepts. Experiments demonstrate that although MLLMs make advances in other scenarios, they exhibit weak ability in association. We speculate that this deficiency may be due to **the lack of learning of unpaired data.** To the best of our knowledge, current MLLMs are trained on image-text pairs and interleaved image-text pair data, providing them with a powerful ability to comprehend input information. However, the association task requires MLLM to have the capability of inference on unpaired sequence data, as well as the ability to gradually uncover the *underlying principles* through the process of all prior decision-making. In the future, an urgent study is still needed to develop a paradigm that links new learning with prior learned knowledge, which may enhance MLLM's association capabilities. We believe that the next stage of MLLMs should expand the learning of unpaired data, which may require the creation of a new learning framework. This advancement will help narrow the gap between MLLMs and human intelligence.

## 7 CONCLUSION

In this paper, we propose a benchmark to evaluate the association ability of MLLMs via an annotation-free association construction method that easily transfers general datasets for association tasks. Using this method, we devise a standard benchmark based on adjective and verb semantic concepts as the association chain. Expanding experiments demonstrate that current open-source MLLMs and even GPT-4V have a significant gap compared to humans.

ACKNOWLEDGMENTS

This work is supported in part by the National Natural Science Foundation of China under Grant No.62306175, CCF-Tencent Rhino-Bird Open Research Fund.

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

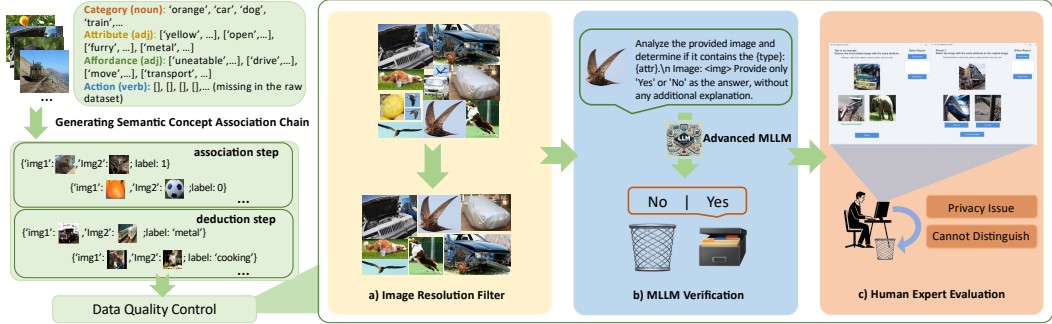

Figure 7: Annotation-free construction method. The left transfers raw supervised data for the association dataset. The right is the expansion of three strict data quality control steps, including image resolution filter, MLLM verification, and human expert evaluation.

# A  IMPLEMENTATION DETAILS

## A.1  IMPLEMENTATION OF DATA REFINEMENT

In our experiment, we comprehensively develop three steps to consolidate the data quality in our tasks, including Image resolution filter, MLLM verification, and Human expert evaluation, as seen in Figure 7.

**Image Resolution Filter.**  In this work, we concentrate on the object within the image, yet the current object detection and object concept learning datasets typically represent only partial content of the images, posing significant perception challenges. Hence, the Image resolution filtering step excludes all images with fewer than 50,000 pixels, ensuring sufficient visual quality.

**MLLM Verification.**  While an image resolution filter ensures that images contain sufficient information for downstream tasks, there remain existing some concerns. One is the correctness of raw annotations, and the other is perception ability in difficult categories. We propose an MLLM verification step that further filters input samples with erroneous annotations or those requiring advanced perception capabilities. This step does not introduce bias, as the filtering method relies on a widely used public dataset with only insignificant shortcomings. Specifically, we employ a question-answers strategy to check whether the model identifies the existence of one concept. This process begins with Gemini-1.5-Flash, then GPT4-V once Gemini-1.5-Flash is unable to reach a decision.

**Human Expert Evaluation.**  The first two steps ensure the correctness of our benchmark, which reduces the image with low quality or confusing annotation. We continue to develop an online testing interface for human testers, as shown in Figure 8, which further filters the images with potential confusion or ethical concerns from the human perspective. We have strictly followed the ethical review, which is described in section H of the supplementary.

## A.2  IMPLEMENTATION OF MEMORY BASE

In our proposed association tasks, whether involving single-step, synchronous and asynchronous associations, we developed three memory strategies to emulate human intelligence: Structure Memory (StructM), Natural Language Memory (NLM), and Chain Memory (ChainM). As in Table 3, the prompt in the association task can be divided into three parts. The first component is the memory context, including the memory instructions and base. Next is the question content, comprising the question instructions and the questions themselves. Finally, the output instructions guide the model to ensure the generated output aligns with our specified requirements.

In detail, for each association step, NoM operates without any memory content. StructM employs simple memory instructions that incorporate memory and structured dictionary knowledge from

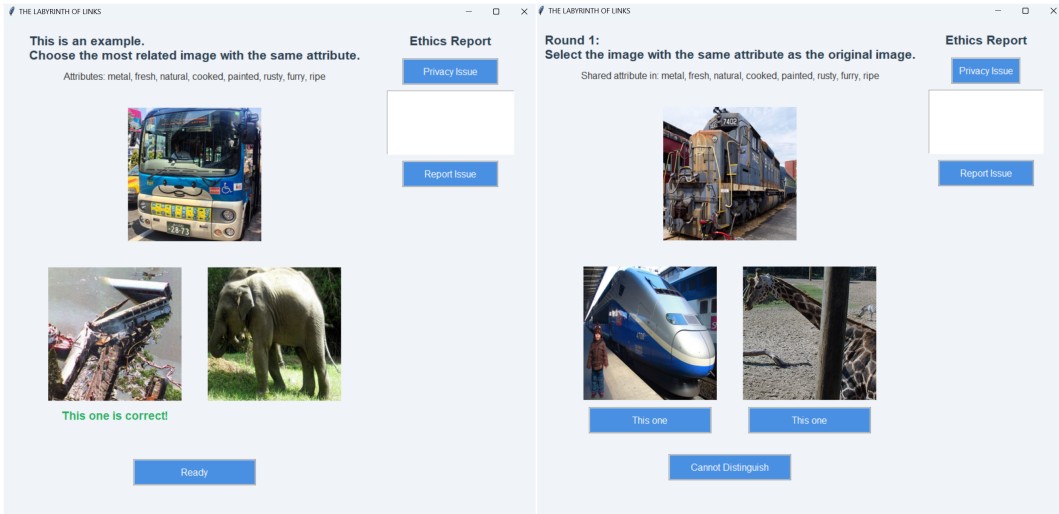

Figure 8: The human evaluation interface in data refinement is designed to filter low-quality data and flag potential ethical issues. The evaluation process consists of two stages: first, a preview phase to familiarize evaluators with sample data (Left), followed by the main testing phase (Right).

| Type | Memory Type | Memory Instruction | Memory base | Question Instruction | Question | Output Instruction |
|---|---|---|---|---|---|---|
| Association Step | NoM | - | - | Determine the relationship between the original image and the candidate images, and select the images with the same attribute/affordance/action as the original image. | Original image:<image>. Candidate images: Image1:<image>, Image2:<image>. | Your response should be direct and exclusively only include one of the following items. Options: [Image1, Image2]. |
| | StructM | Given the memory: | {'eat': ['sandwich', 'pizza']} | | | |
| | NLM | Before this question, you have learnt that related pictures may have the following affordance: | ['broccoli', 'orange'] have eat affordance | | | |
| | ChainM | | broccoli− >eat− >pizza − >eat− >baked | | | |
| Deduction Step | - | - | - | Generate the common affordance between the original image and selected images. | Original image:<image>. Selected image: <image>. | Your response should only include shared affordance in the following options. Options:['break', 'carry', 'clean', 'cut', 'open', 'push', 'sit', 'imprint'] |

Table 3: The prompt format in our association benchmark, which includes the association step and deduction step. The association step makes the selection according to current observations and prior practice. We devise three memory strategies in the association step. In the deduction step, MLLM deducts the underlying concept with the original image and correctly selected image.

prior tasks. Both NLM and ChainM utilize more detailed memory instructions, with NLM representing prior knowledge in everyday language, while ChainM organizes the memory as a task-oriented sequence. Furthermore, the deduction step includes a detailed question description and corresponding output instructions, without relying on any memory content.

In the experiment, we set the repetition weight $w_r$ and forgetting decrement $d_f$ are 1.0 and 0.2 for memory base attention in StructM and NLM in all cases, respectively.

## A.3 IMPLEMENTATION OF EVALUATION SETTING

Transferring from the single-step to synchronous and asynchronous steps, we evaluate the association capability in a dynamic sequential environment. This involves iteratively processing the association step and deduction step and exiting when the association makes an error. In this setting, we evaluate the max / mean step, which is the maximum step and average step across multiple round tests, respectively. The related evaluation setting can refer to Algorithm 1.

To save memory during synchronous and asynchronous steps, we developed three distinct strategies for transferring experiences in the deduction step to update association memory. These strategies encompass memory storage, deep memory retention, and forgetting mechanisms.

---

**Algorithm 1** Synchronous/Asynchronous Association Evaluation

---

1: **procedure** MAIN
2:     **for all** attr $\in$ options **do**
3:         **for** epoch $\leftarrow 1$ **to** $N$ **do**
4:             Reset_Memory()
5:             **for** i $\leftarrow 1$ **to** max_len **do**
6:                 **if** Asynchronous **and** i **mod** $5 = 4$ **then**
7:                     Switch_Attribute()            $\triangleright$ Change to the other attribute in Asynchronous.
8:                 **end if**
9:                 query_img $\leftarrow$ previous_correct_img
10:                correct_img, false_img $\leftarrow$ Get_Next_Images(attr)
11:                correct_answer $\leftarrow$ Random_Order()
                                           $\triangleright$ Randomly set Img1 or Img2 as correct image
12:                response $\leftarrow$ Query_MLLM(memory, query_img, correct_img, false_img)
13:                **if** response $\neq$ correct_answer **then**
14:                    Record_Step_Length(i)
15:                    **break**
16:                **end if**
17:                **if** mode $\neq$ "nomem" **then**
18:                    Deduction_Step(query_img, correct_img)
19:                **end if**
20:             **end for**
21:         **end for**
22:         Calculate_Statistics(steps)          $\triangleright$ Calculate $Max|Mean$ step for all epochs.
23:     **end for**
24: **end procedure**

---

## B  DEEPLY COMPARISON WITH EXISTING BENCHMARK

Our paper is the first to conduct a comprehensive investigation into sequential concept association in MLLMs. - As summarized in a recent benchmark survey Li et al. (2024b), MLLMs' benchmark mainly concentrates on the design of more complex tasks and evaluating nuanced correlations between input samples. - More deep comparison, our benchmark also needs general perception capability as LLaVA-Bench Liu et al. (2024b), nuanced features within images as Compbench Kil et al. (2024), and cooperation across different modalities as SpatialRGPT Cheng et al. (2024). Furthermore, our benchmark needs exceptional abilities beyond existing work. First, our benchmark builds on adjectives and verbs that need deeper perception than nouns in MMVP Tong et al. (2024). Second, our benchmark is sequential images beyond the two images in the existing work of MILEBENCH Song et al. (2024). Finally, our benchmark breaks the closed reasoning into open scenarios, which evaluate an inductive and deductive process rather than ground-truth 'yes/no' in existing work as MARVEL Jiang et al. (2024).

In summary, our benchmark involves the basic capability existing in the previous benchmark, including general perception, nuance features, and cooperation across different modalities. Simultaneously, our benchmark involves exceptional abilities, including deep concept perception, sequential image tasks, and larger solution space.

## C  DEEP ANALYSIS OF MLLM INFERIOR HUMANS

We speculate three possible reasons contributing to the significant gap between MLLMs and humans. The related content can be referred to Figure 10.

**The benchmark needs more precise locationality.** Compared to existing methods Wu et al. (2024) that rely on language questions with explicit information to retrieve related image content, our benchmark includes a comprehensive memory that stores structured knowledge from prior experiences. The key information relevant to the model's predictions is often distributed across different

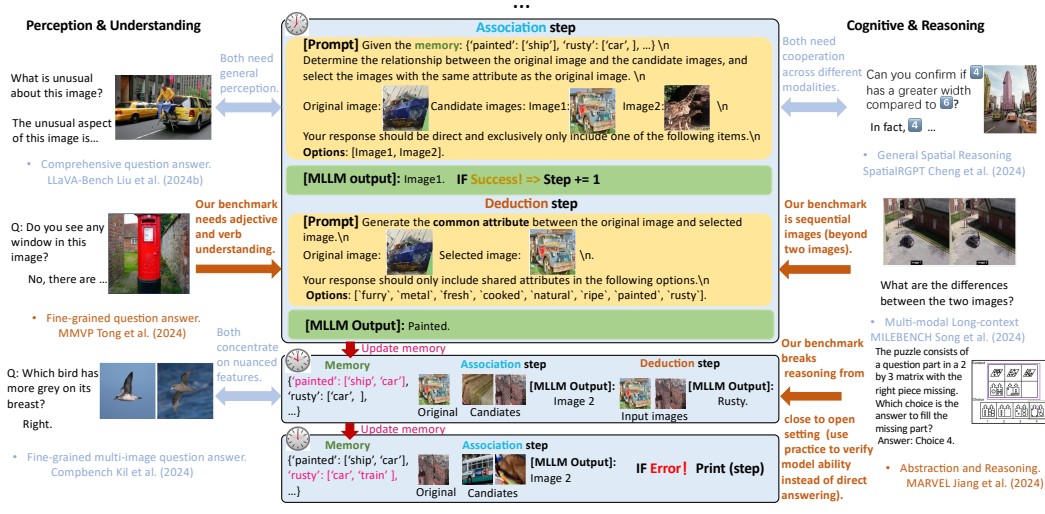

Figure 9: Comparison with existing work. Existing work can be roughly divided into perception & understanding and cognitive & reasoning. Those all target the design of more complex tasks or nuanced evaluation. Conversely, our work concentrates on evaluating sequential concept association between different objects.

locations and represented as relatively local, subtle elements within the context. This highlights the need for more advanced attention in our benchmark.

**The association needs more common sense-based reasoning.** The association is an implicit link between the different objects that are not explicitly shown in the query, requiring a deeper common sense understanding. For instance, existing benchmark Song et al. (2024) infer the weather in the image, they directly use the word in the query to retrieve the part in the image. In contrast, our benchmark only specifies the direction of the link, requiring the model to think of the underlying link within the prompted scope.

**The association needs short-term memory, which is underexplored by existing work.** Our association benchmark is sequential concept links between different objects, which inherently need the memory mechanism to retain the prior experiences. This aspect of memory remains underexplored in existing benchmarks, with current designs falling significantly behind the sophistication of human cognitive systems. This inspires the future study in memory design.

## D PRELIMINARY INVESTIGATION ON CONCEPT PERCEPTION

To thoroughly access the MLLM's ability in concept perception, we systematically design multiple task complexities across multiple tunning-free methods. Specifically, we involve four task complexity settings across two types in all models, including individual and combination perceptions. The individual perception includes Task Instruction (TaskInstr) and Meta Instruction (MetaInstr). However, combination perception consists of Task Instruction with Deduction (TaskInstr w/ DedPo) and Meta Instruction with Deduction (MetaInstr w/ DedPo), which involves another deduction step compared with individual perceptions. The distinction between TaskInstr and MetaInstr lies in whether the prompt includes question instruction that describes the decision-making explanations of the task. In addition, with Deduction (w/ DedPo) compared to normal, is whether we only access perception or evaluate deduction ability. On the other hand, enhancing perception capabilities can be approached from several directions, including expanding foundational knowledge and incorporating practical examples. Hence, we include several prompt engineering skills for tunning-free methods such as common knowledge (ComKnow), one-shot, and Chain-of-Thought (CoT). ComKnow aims to provide foundational knowledge of object-related concepts, while One-shot and CoT focus

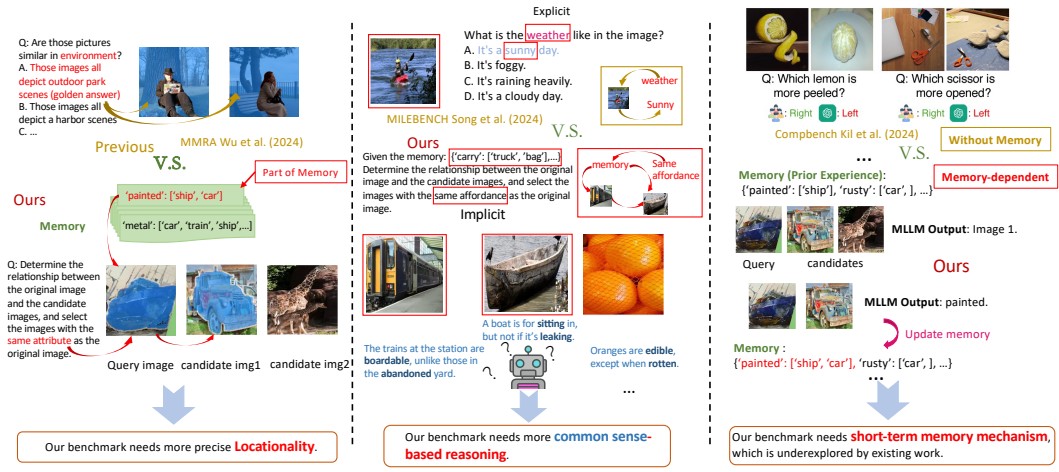

Figure 10: Deep analysis of MLLM inferior humans. We speculate there exist three possible reasons for this gap, including salience attention, common sense-based reasoning, and human-like memory.

on enhancing performance through few-shot learning. Compared to One-shot, CoT introduces an additional reasoning step to emulate human-like thought processes.

We carry out experiments on three open-source MLLMs. QWen-VL Bai et al. (2023b) uses QWen-7B Bai et al. (2023a) as the initialize of LLM and OpenCLIP's ViT-bigG Cherti et al. (2023) as the visual encoder. LLaVA-NeXT-7B Liu et al. (2024a) is constructed on the LLM of Mistral-7B Jiang et al. (2023) and vision encoder of CLIP's ViT-L/14 Radford et al. (2021), while LLaVA-NeXT-13B uses the same vision encoder but is constructed on LLM of Vicuna-13B Zheng et al. (2024).

| Model | Strategy | Concept Perception Task Complexity (Success Ratio) | | | |
|---|---|---|---|---|---|
| | | Individual Perception | | Combination Perception | |
| | | TaskInstr | MetaInstr | TaskInstr w/ DedPo | MetaInstr w/ DedPo |
| | Random | 0.5 | 0.5 | 0.25 | 0.25 |
| QWen-VL | Normal | 0.515 | **0.690** | **0.506** | **0.542** |
| | ComKnow | 0.594 | 0.512 | 0.502 | 0.419 |
| | One-shot | **0.610** | 0.638 | 0.425 | 0.436 |
| | CoT | 0.518 | 0.657 | 0.440 | 0.433 |
| LLaVA-NeXT (Mistral-7B) | Normal | 0.494 | 0.565 | 0.433 | 0.415 |
| | ComKnow | 0.568 | 0.498 | 0.411 | 0.374 |
| | One-shot | 0.641 | 0.631 | 0.453 | 0.475 |
| | CoT | **0.664** | **0.667** | **0.494** | **0.519** |
| LLaVA-NeXT (Vicuna-13B) | Normal | 0.510 | 0.587 | 0.278 | 0.420 |
| | ComKnow | 0.597 | 0.532 | 0.408 | 0.381 |
| | One-shot | **0.598** | 0.605 | 0.411 | 0.421 |
| | CoT | 0.590 | **0.613** | **0.427** | **0.430** |

Table 4: The success ratio on OCL for open-source MLLMs at the concept perception. We include prompt-engineer methods ComKnow, One-shot, CoT. The best results are shown in **bold**.

Table 4 summarises the result on the OCL dataset. In all models, we include four different task complexity settings: Task Instruction (TaskInstr), Meta Instruction (MetaInstr), Instruction with Deduction (TaskInstr w/ DedPo), and Meta Instruction with Deduction (Meta w/ DedPo). For comparison, we include the Random and Normal as baselines, as well as general prompt-engineer skills Common Knowledge (Comknow), One-shot, and Chain-of-Thought (CoT).

Open-source MLLMs perform poorly in concept perception, as reflected in Normal having close performance to Random in all cases. The prompt engineering skills show a slight improvement in some cases. The biggest improvements for ComKnow, One-shot, and CoT are 0.130, 0.147, and 0.170, respectively. While they demonstrate improvements in certain cases, they exhibit weak stability. For instance, in the CoT strategy employed in the TaskInstr w/ DedPo setting, the LLaVA-NeXT-13B has shown an improvement of 0.170 w.r.t Normal whereas Qwen-VL exhibits a negative effect. Combining all results, we conclude that current MLLMs have limited capability in this task.

# E DETAILED RESULTS

## E.1 DETAILED SINGLE-STEP ASSOCIATION RESULT

| Type | Model | Memory | Single-Step Attribute Concept Association Chain | | | | | | | | |
| --- | --- | --- | --- | --- | --- | --- | --- | --- | --- | --- | --- |
| | | | furry | metal | fresh | cooked | natural | ripe | painted | rusty | Avg. |
| Association Success Ratio | LLaVA-Onevision | NoM | 57.63 | 75.21 | 75.78 | 76.09 | 51.19 | 86.83 | 88.19 | 93.24 | 75.52 |
| | | StructM | 57.08 | 79.73 | 83.23 | 78.26 | 53.72 | 89.22 | 87.40 | 98.64 | 78.41 |
| | | NLM | 60.15 | 83.52 | 87.58 | 78.26 | 51.82 | 92.22 | 93.70 | 95.94 | 80.40 |
| | | ChainM | 60.90 | 81.80 | 86.96 | 65.22 | 53.41 | 88.62 | 88.98 | 95.95 | 77.73 |
| | QWen2-VL | NoM | 60.08 | 76.64 | 85.09 | 71.74 | 53.88 | 87.43 | 78.74 | 97.30 | 76.36 |
| | | StructM | 61.31 | 87.51 | 83.23 | 78.26 | 55.78 | 79.04 | 88.98 | 87.84 | 77.74 |
| | | NLM | 65.26 | 93.06 | 96.27 | 97.83 | 52.61 | 96.41 | 92.91 | 98.65 | 86.63 |
| | | ChainM | 63.90 | 91.59 | 92.55 | 93.48 | 55.15 | 97.01 | 93.70 | 1.00 | 73.55 |
| | mPLUG-Owl3 | NoM | 52.72 | 49.68 | 63.35 | 45.65 | 49.60 | 59.28 | 56.69 | 55.41 | 54.05 |
| | | StructM | 61.38 | 72.16 | 75.16 | 63.04 | 56.89 | 82.63 | 79.53 | 90.54 | 72.67 |
| | | NLM | 64.71 | 66.75 | 86.34 | 71.74 | 51.19 | 83.83 | 75.59 | 86.49 | 73.33 |
| | | ChainM | 61.72 | 62.42 | 83.23 | 65.22 | 57.05 | 83.23 | 62.20 | 90.54 | 70.70 |
| | **Gemini-1.5-Flash** | NoM | 59.11 | 71.01 | 82.91 | 84.78 | 55.85 | 89.02 | 68.91 | 91.66 | 75.40 |
| | | StructM | 68.80 | 91.80 | 93.70 | 97.80 | **58.40** | 92.70 | 95.20 | **100.0** | 87.30 |
| | | NLM | 74.90 | 92.90 | 96.80 | 95.50 | 57.50 | 94.80 | 93.90 | 100.0 | 88.39 |
| | | ChainM | 70.30 | 92.60 | 87.10 | 90.90 | 56.40 | 79.90 | 94.40 | 98.6 | 83.78 |
| | **GPT-4o** | NoM | 58.06 | 89.07 | 96.13 | 90.47 | 52.67 | 95.73 | 95.12 | 98.65 | 84.49 |
| | | StructM | 74.11 | **98.98** | **99.39** | 97.83 | 44.95 | **99.40** | 96.00 | 100.0 | 88.83 |
| | | NLM | **78.35** | 96.98 | 96.20 | 97.83 | 53.77 | 95.83 | **100.0** | 100.0 | **89.87** |
| | | ChainM | 72.13 | 90.59 | 81.82 | **100.0** | 46.15 | 81.48 | 66.67 | 100.0 | 79.86 |
| Deduction Success Ratio | LLaVA-Onevision | - | 71.73 | 39.40 | 38.51 | 69.57 | 17.43 | 55.69 | 90.55 | 12.16 | 49.38 |
| | QWen2-VL | - | 63.10 | 69.75 | 51.55 | 63.04 | 39.62 | **59.88** | 59.84 | 58.11 | 58.11 |
| | mPLUG-Owl3 | - | 66.76 | 44.12 | 61.49 | **84.78** | 32.49 | 47.31 | 71.65 | 81.08 | 61.21 |
| | **Gemini-1.5-Flash** | - | **74.44** | 62.18 | 79.75 | 84.78 | 52.10 | 6.70 | **91.60** | 75.00 | 65.82 |
| | **GPT-4o** | - | 59.39 | 86.34 | 94.19 | 73.81 | 90.77 | 52.80 | 86.99 | 80.82 | **78.14** |

Table 5: The success ratio of single-step attribute association, which includes the association step and deduction step. The association step evaluates MLLM's capabilities in decision-making based on observation and corrected prior practice. The deduction step accesses MLLM's capability to generate the common concepts that serve as the underlying rule in the association step. Additionally, the association step includes three memory strategies and a baseline NoM. The best and second results are shown in **bold** and underline, respectively.

In the main text, we highlight three key points. First, the best performance of the closed-source MLLM, Gemini-1.5-Flash, still shows a significant gap compared to human experts. Second, our proposed memory strategies result in noticeable improvements across all cases. Finally, the NLM strategy demonstrates the highest performance compared to our proposed other memory strategies. In this part, we continue to analyze the differences within each concept.

Table 5, 6, 7 summarise the results of single-step association on attribute, affordance, and action, respectively. For each concept type, we assess the association and deduction success rates using three open-source MLLMs—LLaVA-OneVision, QWen2-VL, and mPLUG-Owl3—as well as one closed-source MLLM, Gemini-1.5-Flash. These evaluations are conducted across three memory strategies—StructM, NLM, and ChainM—along with a baseline, NoM. We have selected eight categories for each concept, and provide a detailed analysis of the results below.

| Type | Model | Memory | Single-Step Affordance Concept Association Chain | | | | | | | | |
| --- | --- | --- | --- | --- | --- | --- | --- | --- | --- | --- | --- |
| | | | sit | imprint | push | carry | cut | clean | open | break | Avg. |
| Association Success Ratio | LLaVA-Onevision | NoM | 77.06 | 66.74 | 80.29 | 58.73 | 83.33 | 54.83 | 78.30 | 65.57 | 70.61 |
| | | StructM | 81.18 | 73.43 | 85.53 | 62.20 | 80.44 | 48.39 | 83.96 | 71.99 | 73.39 |
| | | NLM | 87.65 | 75.10 | 87.16 | 60.35 | 83.88 | 58.06 | 88.68 | 68.17 | 76.13 |
| | | ChainM | 86.47 | 72.80 | 83.73 | 60.81 | 82.09 | 53.22 | 90.57 | 68.58 | 74.38 |
| | QWen2-VL | NoM | 81.76 | 67.78 | 79.39 | 62.54 | 85.54 | **72.58** | 81.13 | 70.08 | 75.10 |
| | | StructM | 83.53 | 70.50 | 78.66 | 73.99 | 81.13 | 61.29 | 80.19 | 81.83 | 75.39 |
| | | NLM | 88.82 | 74.90 | 90.96 | 69.25 | 89.67 | 48.39 | 89.62 | 83.33 | 79.37 |
| | | ChainM | 87.65 | 80.96 | 90.24 | 73.53 | 92.42 | 56.45 | 96.23 | 83.88 | 82.67 |
| | mPLUG-Owl3 | NoM | 58.24 | 48.33 | 53.71 | 54.57 | 59.64 | 53.23 | 51.89 | 50.82 | 53.80 |
| | | StructM | 81.76 | 61.51 | 74.86 | 68.09 | 74.24 | 66.13 | 83.96 | 73.63 | 73.02 |
| | | NLM | 78.82 | 61.72 | 67.99 | 60.81 | 79.89 | 46.77 | 78.30 | 68.03 | 67.79 |
| | | ChainM | 87.18 | 61.72 | 64.20 | 62.77 | 75.62 | 59.68 | 67.92 | 68.58 | 68.46 |
| | **Gemini-1.5-Flash** | NoM | 86.39 | 66.67 | 83.33 | 62.15 | 95.18 | 63.93 | 83.81 | 74.79 | 77.78 |
| | | StructM | 97.02 | 70.72 | 91.77 | 73.99 | 98.19 | 58.33 | **99.04** | 86.55 | 84.45 |
| | | NLM | 97.02 | 83.64 | 92.90 | 74.79 | 98.20 | 55.17 | 98.06 | 83.44 | 85.40 |
| | | ChainM | 94.64 | 76.50 | 86.29 | 66.40 | 95.58 | 46.67 | 95.19 | 78.92 | 80.02 |
| | **GPT-4o** | NoM | 88.54 | 77.58 | 92.47 | 75.64 | 98.96 | 60.98 | 95.92 | **89.68** | 84.93 |
| | | StructM | **98.21** | **87.05** | 94.97 | 86.46 | 74.59 | 59.02 | 98.11 | 84.46 | 85.36 |
| | | NLM | 97.65 | 80.28 | 96.97 | 86.76 | 100.0 | 55.56 | 96.00 | 80.82 | 86.76 |
| | | ChainM | 87.50 | 78.26 | **100.0** | 83.33 | 97.99 | 58.06 | 68.86 | 78.28 | 81.54 |
| Deduction Success Ratio | LLaVA-Onevision | - | 66.47 | 0.42 | 8.14 | 1.04 | 4.96 | **35.48** | 49.06 | 3.01 | 21.07 |
| | QWen2-VL | - | 21.76 | **14.23** | 1.63 | 1.50 | 1.24 | 4.84 | 80.19 | 2.46 | 15.98 |
| | mPLUG-Owl3 | - | **82.94** | 0.84 | 12.12 | 2.89 | 63.91 | 12.90 | 84.91 | 5.87 | 33.30 |
| | **Gemini-1.5-Flash** | - | 58.93 | 0.85 | 24.40 | 36.02 | 53.01 | 5.00 | 57.69 | **28.71** | 33.08 |
| | **GPT-4o** | - | 63.54 | 1.75 | 10.22 | 21.94 | **71.88** | 17.50 | 27.55 | 4.05 | 27.30 |

Table 6: Same as Table 5, but for affordance single-step association.

**Comparison of Different Models.** From the whole perspective, Gemini-1.5-Flash consistently outperforms the other models across all cases. Specifically, for association success ratio, Qwen2-VL leads in several categories of each concept. For instance, metal and cooked with NLM strategy attain the highest success ratio in attribute single-step association. Among the open-source MLLMs, Qwen2-VL demonstrates the strongest performance across three concepts, with LLaVA-OneVision performing moderately well but trailing behind Qwen2-VL, and mPLUG-Owl3 showing the lowest performance. Furthermore, different MLLMs exhibit consistent trends in concept understanding. For instance, LLaVA-OneVision underperforms in natural attribute association, the same pattern is also observed with QWen2-VL and mPLUG-Owl3 models. Additionally, there is a significant disparity in deduction success rates among the different MLLMs. For example, Gemini-1.5-Flash performs well with "push", "carry", and "break" affordances, while the other MLLMs show relatively weaker capabilities. Conversely, Qwen2-VL excels in "imprint" affordance, and LLaVA-OneVision demonstrates strength in the 'clean' affordance, whereas Gemini-1.5-Flash struggles.

**Comparison of Different Categories.** Although single-step association shows consistent improvements across various concepts, performance differences emerge in specific categories. For instance, fresh, ripe, and rusty attributes gain an association success ratio close to 1, but natural and ripe attributes with a lower performance compared to the random success ratio of 0.5. This phenomenon is more pronounced in the deduction success ratio. Several model implementations exhibit strong deduction capabilities across different categories. For instance, Gemini-1.5-Flash achieves a 91.6 deduction success ratio in the 'painted' attribute. However, some categories still demonstrate lower performance, highlighting insufficient object understanding.

**Comparison of Different Memory Strategies.** As demonstrated in other subsection analyses, NLM exhibits an overall more powerful performance than StructM and ChainM. But there also exists some observations that violate this finding, which further highlight the instability of the current MLLM. For instance, in the case of Gemini-1.5-Flash with "natural" and "painted" attributes, the success ratio of StructM strategy outperforms NLM. They demonstrate professional-level ability in

| Type | Model | Memory | Single-Step Action Concept Association Chain | | | | | | | | |
|---|---|---|---|---|---|---|---|---|---|---|---|
| | | | run | hit | drive | dress | cooking | build | shake | cut | Avg. |
| Association Success Ratio | LLaVA-OneVision | NoM | 74.91 | 73.25 | 79.48 | 75.36 | 88.15 | 74.05 | 82.21 | 58.53 | 75.74 |
| | | StructM | 72.40 | 77.23 | 77.65 | 75.36 | 85.63 | 78.61 | 79.95 | 56.67 | 75.44 |
| | | NLM | 78.10 | 77.50 | 82.50 | 75.36 | 88.03 | 79.99 | 87.39 | 60.44 | 78.66 |
| | | ChainM | 73.89 | 74.49 | 80.53 | 75.36 | 87.13 | 79.56 | 85.36 | 59.01 | 76.92 |
| | QWen2-VL | NoM | 83.32 | 80.52 | 82.50 | 65.22 | 87.70 | 74.68 | 87.61 | 65.89 | 78.43 |
| | | StructM | 80.71 | 79.29 | 81.01 | 75.36 | 87.89 | 87.22 | 94.82 | 70.50 | 82.10 |
| | | NLM | 89.72 | 86.97 | 88.66 | 79.71 | 95.31 | 91.87 | 96.17 | 75.70 | 88.01 |
| | | ChainM | 85.99 | 85.05 | 82.87 | 76.81 | 93.13 | 87.08 | 97.52 | **76.24** | 85.59 |
| | mPLUG-Owl3 | NoM | 56.10 | 52.26 | 59.02 | 53.62 | 66.49 | 63.34 | 57.43 | 55.24 | 57.94 |
| | | StructM | 71.28 | 69.55 | 80.13 | 60.87 | 86.30 | 77.49 | 84.68 | 61.04 | 73.92 |
| | | NLM | 63.19 | 62.41 | 76.45 | 63.77 | 78.15 | 73.24 | 81.98 | 61.16 | 70.04 |
| | | ChainM | 64.04 | 62.14 | 78.64 | 50.72 | 82.57 | 73.39 | 84.91 | 59.55 | 69.50 |
| | **Gemini-1.5-Flash** | NoM | 85.95 | 88.96 | 89.23 | 71.88 | 94.16 | 81.94 | 92.05 | 69.47 | 84.21 |
| | | StructM | 86.49 | 89.81 | 85.71 | 77.97 | **99.60** | 93.20 | 97.72 | 74.33 | 88.10 |
| | | NLM | **93.07** | 92.36 | 92.28 | 75.41 | 98.99 | **93.78** | 97.28 | 73.48 | **89.58** |
| | | ChainM | 86.46 | 93.54 | 88.34 | 74.63 | **99.60** | 86.23 | 98.41 | 73.42 | 87.58 |
| | **GPT-4o** | NoM | 86.75 | 89.24 | **92.86** | 75.56 | 95.34 | 82.56 | 96.95 | 76.51 | 86.97 |
| | | StructM | 81.87 | **96.39** | 91.62 | 73.52 | 99.42 | 88.44 | **100.0** | **78.53** | 88.72 |
| | | NLM | 87.37 | 89.41 | 89.12 | **81.97** | 99.00 | 80.90 | 90.50 | 70.74 | 86.13 |
| | | ChainM | 85.00 | 90.40 | 91.00 | 67.65 | 98.99 | 86.29 | 97.99 | 69.85 | 85.90 |
| Deduction Success Ratio | LLaVA-Onevision | - | 58.87 | 48.97 | 73.35 | 34.78 | 96.85 | 3.96 | 3.60 | 52.48 | 46.61 |
| | QWen2-VL | - | 60.90 | 69.00 | 67.41 | 62.32 | 89.78 | 7.64 | 3.60 | 39.26 | 49.99 |
| | mPLUG-Owl3 | - | 33.99 | 4.94 | 74.52 | 44.93 | 96.74 | 7.72 | 5.63 | **74.62** | 42.89 |
| | **Gemini-1.5-Flash** | - | **67.21** | 18.00 | **86.99** | **79.69** | 97.79 | 19.96 | **59.68** | 16.22 | 55.69 |
| | **GPT-4o** | - | 38.50 | **70.20** | 83.50 | 58.82 | **100.0** | **21.83** | 31.66 | 53.27 | **57.28** |

Table 7: Same as Table 5, but for action single-step association.

specific cases but cannot maintain consistent stability. We speculate that similar to the prominent research area of hallucinations, there is still significant room for improvement.

## E.2 DETAILED SYNCHRONOUS ASSOCIATION RESULT

| Model | Memory | Synchronous Attribute Concept Association Chain (Max \| Mean Step) | | | | | | | | |
|---|---|---|---|---|---|---|---|---|---|---|
| | | furry | metal | fresh | cooked | natural | ripe | painted | rusty | Avg. |
| LLaVA-OneVision | NoM | 15 \| 1.30 | 31 \| 3.20 | 39 \| 3.85 | 38 \| 3.13 | 11 \| 1.06 | 40 \| 5.69 | 70 \| 6.68 | 140 \| 17.47 | 48.0 \| 5.30 |
| | StructM | 15 \| 1.38 | 30 \| 4.03 | 41 \| 4.75 | 32 \| 4.04 | 12 \| 1.21 | 51 \| 6.51 | 65 \| 9.23 | 238 \| 15.24 | 60.6 \| 5.80 |
| | NLM | 14 \| 1.46 | 40 \| 5.24 | 68 \| 6.09 | 48 \| 3.93 | 12 \| 1.16 | 91 \| 8.88 | 81 \| 10.64 | 249 \| 33.88 | 75.4 \| 8.91 |
| | ChainM | 16 \| 1.49 | 63 \| 4.66 | 58 \| 5.23 | 28 \| 3.29 | 11 \| 1.13 | 63 \| 6.98 | 70 \| 9.70 | 177 \| 23.79 | 60.8 \| 7.03 |
| Qwen2-VL | NoM | **21** \| 1.41 | 39 \| 3.14 | 63 \| 8.32 | 43 \| **4.52** | 14 \| 1.16 | 94 \| 10.98 | 57 \| 4.44 | 353 \| 46.05 | 85.5 \| 10.00 |
| | StructM | 13 \| 1.61 | 25 \| 2.36 | 28 \| 3.46 | 23 \| 2.63 | 22 \| **1.34** | 37 \| 3.42 | | 122 \| 9.61 | 37.9 \| 3.57 |
| | NLM | 15 \| **1.74** | 58 \| 6.77 | 89 \| 8.50 | 36 \| **4.52** | 11 \| 1.27 | 109 \| 11.90 | 162 \| **12.50** | 500 \| 64.50 | 122.5 \| **13.96** |
| | ChainM | 18 \| 1.64 | 39 \| 4.38 | 42 \| 7.53 | 32 \| 4.04 | 10 \| 1.18 | 77 \| 9.75 | 68 \| 8.50 | 169 \| 22.06 | 56.9 \| 7.39 |
| mPLUG-Owl3 | NoM | 11 \| 1.06 | 12 \| 1.08 | 17 \| 1.39 | 13 \| 1.27 | 17 \| 1.46 | 13 \| 1.28 | 16 \| 1.54 | | 13.4 \| 1.27 |
| | StructM | 15 \| 1.71 | 24 \| 2.34 | 41 \| 4.52 | 22 \| 2.85 | 12 \| 1.20 | 37 \| 5.07 | 39 \| 4.55 | 85 \| 7.31 | 34.4 \| 3.69 |
| | NLM | 19 \| 1.55 | 17 \| 2.03 | 33 \| 4.16 | 27 \| 2.34 | 16 \| 1.21 | 35 \| 4.48 | 39 \| 3.25 | 86 \| 8.30 | 34.0 \| 3.42 |
| | ChainM | 15 \| 1.59 | 20 \| 1.77 | 43 \| 4.38 | 18 \| 1.92 | 12 \| 1.19 | 51 \| 4.63 | 24 \| 2.65 | 69 \| 8.82 | 31.5 \| 3.37 |
| **Human** | Expert-X | 22 | 121 | 500 | 500 | 10 | 500 | 500 | 500 | 331.6 |
| | Expert-Y | 11 | 500 | 500 | 102 | 17 | 500 | 500 | 500 | 328.8 |
| | Expert-Z | 31 | 500 | 500 | 500 | 23 | 500 | 500 | 500 | 381.8 |
| | Max \| Mean | 31 \| 21.3 | 500 \| 373.7 | 500 \| 500 | 500 \| 367.3 | 23 \| 16.7 | 500 \| 500 | 500 \| 500 | 500 \| 500 | 381.8 \| 347.5 |

Table 8: The Max| Mean Step of synchronous attribute concept association across open-source MLLM and human experts. We select eight categories to reduce the complexity of the association and ensure the data quality. For open-source MLLMs, we involve three memory strategies, StructM, NLM, and ChainM, along with a baseline strategy, NoM. The best and second results are shown in **bold** and underline, respectively.

As summarized in the main text, a significant gap exists between current MLLM and human intelligence in synchronous association. Below, we provide a detailed analysis of individual concept associations across each category, focusing on OCL attributes, affordances, and the actions of Pangea.

| Model | Memory | Synchronous Affordance Concept Association Chain (Max \| Mean Step) | | | | | | | | |
| --- | --- | --- | --- | --- | --- | --- | --- | --- | --- | --- |
| | | sit | imprint | push | carry | cut | clean | open | break | Avg. |
| LLaVA-OneVision | NoM | 40 \| 4.17 | 39 \| 2.21 | 47 \| 4.34 | 14 \| 1.49 | 49 \| 5.63 | 22 \| 1.60 | 53 \| 6.74 | 20 \| 2.06 | 35.5 \| 3.53 |
| | StructM | 38 \| 4.53 | 27 \| 2.82 | 46 \| 5.65 | 22 \| 1.86 | 51 \| 7.41 | 16 \| 2.10 | 73 \| 8.64 | 28 \| 2.86 | 37.6 \| 4.48 |
| | NLM | 50 \| 5.70 | 30 \| 2.88 | 48 \| 6.16 | 26 \| 1.95 | 45 \| 6.73 | 19 \| 2.07 | 95 \| 10.95 | 28 \| 2.61 | 42.6 \| 4.88 |
| | ChainM | 37 \| 4.76 | 34 \| 2.82 | 51 \| 5.77 | 13 \| 1.59 | 57 \| 5.82 | 19 \| 1.93 | 77 \| 8.99 | 20 \| 2.32 | 38.5 \| 4.25 |
| Qwen2-VL | NoM | 36 \| 4.17 | 21 \| 2.76 | 36 \| 3.96 | 15 \| 1.88 | 74 \| 7.15 | 18 \| 1.68 | 39 \| 4.44 | 38 \| 2.76 | 34.6 \| 3.60 |
| | StructM | 51 \| 3.80 | 21 \| 2.29 | 25 \| 2.39 | 19 \| 1.86 | 23 \| 2.65 | 16 \| 1.71 | 36 \| 3.10 | 18 \| 2.17 | 26.1 \| 2.50 |
| | NLM | 48 \| 6.12 | 34 \| 2.75 | 35 \| 4.42 | 25 \| 2.61 | 68 \| 6.94 | 15 \| 1.90 | 52 \| 6.37 | 32 \| 3.57 | 38.6 \| 4.34 |
| | ChainM | 47 \| 6.08 | 22 \| 3.04 | 37 \| 4.81 | 23 \| 2.59 | 59 \| 7.25 | 24 \| 1.99 | 63 \| 7.75 | 25 \| 3.45 | 37.5 \| 4.62 |
| mPLUG-Owl3 | NoM | 14 \| 1.42 | 12 \| 1.00 | 13 \| 1.14 | 14 \| 1.12 | 13 \| 1.26 | 13 \| 1.02 | 10 \| 1.14 | 14 \| 1.28 | 12.9 \| 1.72 |
| | StructM | 27 \| 3.94 | 20 \| 1.83 | 27 \| 3.44 | 21 \| 2.69 | 15 \| 1.85 | 31 \| 3.14 | | 40 \| 5.39 | 29.3 \| 3.44 |
| | NLM | 23 \| 3.18 | 13 \| 1.45 | 21 \| 2.30 | 21 \| 1.88 | 32 \| 3.69 | 13 \| 1.39 | 21 \| 2.24 | 34 \| 3.96 | 22.3 \| 2.51 |
| | ChainM | 29 \| 2.94 | 17 \| 1.56 | 21 \| 2.68 | 24 \| 1.96 | 35 \| 3.79 | 16 \| 1.48 | 19 \| 2.34 | 42 \| 4.25 | 25.4 \| 2.63 |
| **Human** | Expert-X | 77 | 105 | 500 | 500 | 500 | 500 | 500 | 500 | 397.8 |
| | Expert-Y | 49 | 93 | 31 | 500 | 500 | 500 | 136 | 500 | 288.7 |
| | Expert-Z | 42 | 78 | 500 | 500 | 500 | 73 | 500 | 500 | 336.2 |
| | Max \| Mean | 77 \| 56.0 | 105 \| 92.0 | 500 \| 343.7 | 500 \| 500.0 | 500 \| 500.0 | 500 \| 357.6 | 500 \| 378.7 | 500 \| 500.0 | 397.8 \| 341.0 |

Table 9: Same as Table 8, but for synchronous affordance concept association.

| Model | Memory | Synchronous Action Concept Association Chain (Max \| Mean Step) | | | | | | | | |
| --- | --- | --- | --- | --- | --- | --- | --- | --- | --- | --- |
| | | run | hit | drive | dress | cooking | build | shake | cut | Avg. |
| LLaVA-OneVision | NoM | 30 \| 2.86 | 22 \| 3.08 | 31 \| 3.80 | 22 \| 2.09 | 78 \| 7.05 | 29 \| 2.93 | 36 \| 3.55 | 13 \| 1.48 | 32.6 \| 3.35 |
| | StructM | 23 \| 2.57 | 25 \| 2.99 | 32 \| 3.61 | 27 \| 2.52 | 55 \| 6.49 | 30 \| 3.11 | 34 \| 4.71 | 15 \| 1.73 | 30.1 \| 3.47 |
| | NLM | 25 \| 2.95 | 27 \| 3.66 | 32 \| 3.84 | 22 \| 2.36 | 56 \| 6.61 | 29 \| 2.88 | 63 \| 5.86 | 16 \| 1.71 | 33.8 \| 3.73 |
| | ChainM | 22 \| 2.60 | 27 \| 3.05 | 31 \| 3.90 | 25 \| 2.36 | 52 \| 5.37 | 25 \| 2.55 | 43 \| 5.36 | 18 \| 1.69 | 30.4 \| 3.36 |
| Qwen2-VL | NoM | 49 \| 5.40 | 42 \| 5.04 | 48 \| 4.90 | 21 \| 2.37 | 79 \| 7.35 | 35 \| 3.18 | 80 \| 10.85 | 24 \| 1.98 | 47.3 \| 5.13 |
| | StructM | 19 \| 2.57 | 23 \| 2.71 | 30 \| 3.15 | 17 \| 2.31 | 46 \| 5.41 | 31 \| 3.62 | 119 \| 12.56 | 20 \| 2.31 | 38.1 \| 4.33 |
| | NLM | 48 \| 6.27 | 35 \| 4.94 | 64 \| 5.57 | 30 \| 3.76 | 162 \| 16.48 | 49 \| 5.59 | 197 \| 24.07 | 24 \| 2.86 | 76.1 \| 8.69 |
| | ChainM | 51 \| 5.06 | 43 \| 4.85 | 37 \| 5.11 | 26 \| 3.47 | 80 \| 12.09 | 45 \| 4.65 | 163 \| 19.07 | 28 \| 2.64 | 59.1 \| 7.12 |
| mPLUG-Owl3 | NoM | 12 \| 1.23 | 11 \| 1.17 | 14 \| 1.51 | 13 \| 1.14 | 20 \| 2.20 | 15 \| 1.77 | 16 \| 1.26 | 16 \| 1.49 | 14.6 \| 1.47 |
| | StructM | 23 \| 2.20 | 18 \| 2.01 | 24 \| 3.29 | 19 \| 1.97 | 64 \| 6.17 | 34 \| 3.90 | 24 \| 1.95 | 40 \| 5.00 | 30.8 \| 3.31 |
| | NLM | 23 \| 1.70 | 16 \| 1.63 | 24 \| 2.49 | 18 \| 1.41 | 39 \| 3.89 | 30 \| 2.94 | 21 \| 1.98 | 39 \| 4.59 | 26.3 \| 2.58 |
| | ChainM | 13 \| 1.94 | 16 \| 1.71 | 37 \| 3.10 | 14 \| 1.79 | 54 \| 4.61 | 28 \| 3.12 | 22 \| 1.90 | 35 \| 4.17 | 27.4 \| 2.79 |
| **Human** | Expert-X | 14 | 70 | 30 | 17 | 500 | 31 | 77 | 19 | 94.8 |
| | Expert-Y | 22 | 14 | 10 | 10 | 500 | 15 | 26 | 17 | 76.8 |
| | Expert-Z | 7 | 22 | 5 | 37 | 500 | 37 | 31 | 90 | 91.1 |
| | Max \| Mean | 22 \| 14.3 | 70 \| 35.3 | 30 \| 15.0 | 37 \| 21.3 | 500 \| 500.0 | 37 \| 27.7 | 77 \| 44.7 | 90 \| 42.0 | 107.9 \| 87.5 |

Table 10: Same as Table 8, but for synchronous action concept association.

Table 8, 9, 10 shows the results of synchronous concept association on attribute, affordance, and action, respectively. For each concept, we include the same eight categories as in the single-step association. We include three open-source MLLMs in synchronous association, including LLaVA-OneVision, QWen2-VL, and mPLUG-Owl3. Additionally, we also integrate this setting into the human test interface. In this setting, we use Max | Mean Step as the primary metric.

**Comparison of Different Models and Memory Strategies.** We find that QWen2-VL in the attribute and action concepts outperforms other models in almost all cases. But, LLaVA-OneVision exhibits extraordinary capabilities in affordance association. This demonstrates the inconsistency of MLLM's concept understanding, which can also taken as a reference for future improvement of MLLM's ability. Interestingly, memory strategies perform worse than NoM in certain instances. For example, in the "cooked" attribute concept association under the QWen2-VL model, NoM achieves a mean-step of $4.52$, while other memory strategies fall below this. We speculate that this attribute may have encountered deduction errors, as shown in Figure 5 in the main text. The more remarkable situation is QWen2-VL with StructM strategy, in which memory prevents the utilization of correct input context. This leads to it attaining low performance compared to NoM.

**Comparison of Different Categories.** Notably, the MLLM demonstrates a similar pattern to human intelligence, as both humans and open-source MLLMs achieve low mean-step scores in the "furry" and "natural" attributes, as shown in Table 8. More specifically, different categories within each concept exhibit different capabilities, which are also inconsistent with human experts. For instance, in the synchronous association of individual affordances, human experts take an average of fewer than 100 steps to complete the "sit" affordance, which presents certain challenges for testers. In comparison, QWen2-VL achieves an average of 6.12 steps, though this exceeds the steps required

for the "carry" affordance. Notably, human experts are less likely to make errors when performing the "carry" affordance.

### E.3   DETAILED ASYNCHRONOUS ASSOCIATION RESULT

| Model | Memory | Asynchronous Attribute Concept Association Chain (Max \| Mean Step) | | | | |
|---|---|---|---|---|---|---|
| | | furry-metal | fresh-cooked | natural-ripe | painted-rusty | Avg. |
| LLaVA-OneVision | NoM | 11 \| 1.15 | 20 \| 3.07 | 18 \| 0.99 | 43 \| 5.39 | 23.0 \| 2.65 |
| | StructM | 10 \| 1.24 | 36 \| 3.52 | 17 \| 1.11 | 59 \| 6.63 | 30.5 \| 3.13 |
| | NLM | 17 \| 1.36 | 32 \| 4.24 | 15 \| 1.19 | 83 \| 8.97 | 36.8 \| 3.94 |
| | ChainM | 11 \| 1.21 | 32 \| 3.47 | 17 \| 1.08 | 67 \| 7.65 | 31.8 \| 3.35 |
| QWen2-VL | NoM | 13 \| 1.24 | 29 \| 4.47 | 16 \| 1.19 | 44 \| 6.22 | 25.5 \| 3.28 |
| | StructM | 12 \| 1.33 | 19 \| 2.55 | 18 \| 1.43 | 38 \| 3.75 | 21.8 \| 2.26 |
| | NLM | 16 \| 1.63 | 43 \| 4.69 | 19 \| 1.49 | 97 \| 9.91 | 43.8 \| 4.43 |
| | ChainM | 10 \| 1.42 | 32 \| 4.50 | 16 \| 1.19 | 56 \| 6.67 | 28.5 \| 3.45 |
| mPLUG-Owl3 | No | 10 \| 1.03 | 14 \| 1.26 | 13 \| 1.04 | 16 \| 1.23 | 13.3 \| 1.14 |
| | StructM | 11 \| 1.44 | 34 \| 3.26 | 16 \| 1.16 | 39 \| 4.28 | 25.0 \| 2.54 |
| | NLM | 12 \| 1.46 | 24 \| 3.07 | 11 \| 1.25 | 25 \| 2.62 | 18.0 \| 2.10 |
| | ChainM | 12 \| 1.32 | 23 \| 2.68 | **22** \| 1.13 | 24 \| 2.77 | 20.3 \| 1.98 |
| **MoE** | No | 8 \| 1.11 | 28 \| 4.06 | 19 \| 1.05 | 56 \| 7.05 | 27.8 \| 3.32 |
| | StructM | 13 \| 1.39 | 49 \| 3.93 | 13 \| 1.25 | 96 \| 8.34 | 42.8 \| 3.73 |
| | NLM | **19** \| 1.53 | 39 \| 5.26 | 17 \| 1.26 | 137 \| 15.91 | 53.0 \| 5.99 |
| | ChainM | 12 \| 1.36 | 39 \| 4.40 | 12 \| 1.17 | 82 \| 11.38 | 36.3 \| 4.58 |
| **Gemini-1.5-Flash** | No | 7 \| 1.68 | 18 \| 5.38 | 8 \| 1.84 | 15 \| 3.35 | 12.0 \| 3.06 |
| | StructM | 9 \| 2.46 | **76** \| **13.8** | 6 \| 2.46 | 186 \| 49.5 | 69.3 \| 17.1 |
| | NLM | 9 \| 2.80 | 31 \| 8.85 | 8 \| 2.30 | **305** \| **97.3** | **88.3** \| **27.8** |
| | ChainM | 8 \| 1.53 | 24 \| 5.30 | 8 \| 2.43 | 51 \| 14.9 | 22.8 \| 6.04 |
| **GPT-4V** | No | 7 \| 2.07 | 53 \| 12.1 | 8 \| 2.93 | 121 \| 24.3 | 47.3 \| 10.4 |
| | StructM | 15 \| **4.33** | 39 \| 8.63 | 6 \| **3.10** | 151 \| 37.3 | 52.8 \| 13.3 |
| | NLM | 8 \| 3.07 | 53 \| 10.5 | 7 \| 2.33 | 168 \| 45.5 | 59.0 \| 15.4 |
| | ChainM | 7 \| 2.35 | 54 \| 9.70 | 7 \| 1.95 | 122 \| 38.2 | 47.5 \| 13.1 |
| **Human** | Expert-X | 16 | 70 | 16 | 500 | 150.5 |
| | Expert-Y | 21 | 54 | 80 | 500 | 163.8 |
| | Expert-Z | 79 | 42 | 11 | 295 | 106.8 |
| | Max \| Mean | 79 \| 38.7 | 70 \| 55.3 | 80 \| 35.7 | 500 \| 431 | 182.3 \| 140.2 |

Table 11: The Max | Mean step on paired attribute categories in asynchronous association setting. We involve three open-source MLLMs, a cutting-edged MoE based on open-source MLLMs, two close-sourced MLLMs, and human experts. In open-source and close-source MLLM, we involve three memory strategies, SturctM, NLM, and ChainM, as well as one baseline strategy NoM. The best and second results are shown in **bold** and underline, respectively.

Paired Concept Association is the most complex task in our work. We find that all max-step and mean-step decrease in this setting compared to individual concepts association in the main text. In this subsection, we detailed analysis of the paired categories association on adjectives and verb concepts, *i.e.*, attribute and affordance in OCL, and action in Pangea.

Table 11, 12, 13 summarise the results of paired categories concept association on attribute, affordance and action respectively. In this section, we include the closed-source MLLMs Gemini-1.5-Flash and GPT-4V, along with the open-source MLLMs LLaVA-OneVision, QWen2-VL, and mPLUG-Owl3, as well as an MoE that integrates all open-source MLLMs and human expertise. We include eight categories from each concept of attribute, affordance, and action.

**Comparison of Different Models and Memory Strategies.**   All the max and mean steps decrease on each paired category by comparing with individual concept association, which is consistence with the whole results. We speculate that this stems from a weak ability to transfer knowledge between concepts, as human testers also require some time to adjust to new rules. Additionally, MoE outperforms open-source MLLMs in some paired categories, where there also exists some cases in which MoE attains a balanced performance compared to one of the highest MLLMs. For instance, for paired categories of "sit-imprint" and "open-break", MoE outperforms all open-source MLLM in all cases, while "push-carry" and "cut-clean" open-source MLLM leads the performance when comparing the open-source setting. On the other hand, different memory strategies remain

| Model | Memory | Asynchronous Affordance Concept Association Chain (Max \| Mean Step) | | | | |
| --- | --- | --- | --- | --- | --- | --- |
| | | sit-imprint | push-carry | cut-clean | open-break | Avg. |
| LLaVA-OneVision | NoM | 29 \| 3.13 | **31** \| **3.26** | 18 \| 3.02 | 15 \| 3.02 | 23.3 \| 3.11 |
| | StructM | **41** \| 3.42 | 13 \| 2.41 | **22** \| 2.92 | 29 \| 3.82 | 26.3 \| 3.14 |
| | NLM | 34 \| 4.09 | 13 \| 2.70 | 18 \| 3.45 | 21 \| 4.12 | 21.5 \| 3.59 |
| | ChainM | 37 \| 3.56 | 12 \| 2.65 | 15 \| 3.19 | 21 \| 3.77 | 21.3 \| 3.29 |
| QWen2-VL | NoM | 26 \| 3.30 | 16 \| 2.58 | 22 \| 3.70 | 19 \| 2.93 | 20.8 \| 3.13 |
| | StructM | 28 \| 2.58 | 17 \| 1.89 | 18 \| 2.13 | 17 \| 2.51 | 20 \| 2.28 |
| | NLM-3 | 29 \| 4.05 | 16 \| 2.98 | 20 \| 3.29 | 28 \| 4.18 | 23.3 \| 3.63 |
| | ChainM | 25 \| 3.54 | 14 \| 2.79 | 20 \| 3.55 | 32 \| 3.93 | 22.8 \| 3.45 |
| mPLUG-Owl3 | NoM | 13 \| 1.31 | 12 \| 1.10 | 14 \| 1.23 | 10 \| 1.27 | 12.3 \| 1.23 |
| | StructM | 20 \| 2.75 | 22 \| 1.90 | 21 \| 3.43 | 21 \| 3.19 | 21.0 \| 2.82 |
| | NLM | 15 \| 2.19 | 14 \| 1.59 | 20 \| 2.64 | 17 \| 2.56 | 16.5 \| 2.25 |
| | ChainM | 15 \| 2.19 | 16 \| 2.05 | 20 \| 3.08 | 16 \| 2.70 | 16.8 \| 2.51 |
| **MoE** | NoM | 26 \| 3.08 | 19 \| 2.44 | 21 \| 3.10 | 15 \| 3.25 | 20.3 \| 2.97 |
| | StructM | 27 \| 3.40 | 14 \| 2.41 | 18 \| 3.16 | 23 \| 3.84 | 20.5 \| 3.20 |
| | NLM | 38 \| **6.41** | 20 \| 2.93 | 16 \| 3.66 | 24 \| 4.42 | 24.5 \| 4.36 |
| | ChainM | 36 \| 3.65 | 16 \| 2.95 | 20 \| 3.62 | 21 \| 4.17 | 23.4 \| 3.60 |
| **Gemini-1.5-Flash** | NoM | 26 \| 3.36 | 11 \| 2.56 | 14 \| 3.12 | 14 \| 4.26 | 16.3 \| 3.33 |
| | StructM | 32 \| 4.67 | 8 \| 2.37 | 19 \| 3.73 | **68** \| **6.93** | **31.8** \| **4.43** |
| | NLM | 13 \| 2.90 | 7 \| 2.30 | 8 \| 3.07 | 24 \| 5.43 | 13.0 \| 3.43 |
| | ChainM | 11 \| 3.67 | 10 \| 2.93 | 9 \| 2.10 | 15 \| 3.27 | 11.3 \| 2.99 |
| **GPT-4V** | NoM | 17 \| 6.00 | 14 \| 2.67 | 14 \| 3.43 | 8 \| 3.00 | 13.3 \| 3.78 |
| | StructM | 29 \| 4.70 | 7 \| 1.70 | 13 \| **4.00** | 19 \| 4.61 | 17.0 \| 3.75 |
| | NLM | 23 \| 6.17 | 9 \| 2.21 | 14 \| 3.57 | 24 \| 4.45 | 17.5 \| 4.10 |
| | ChainM | 16 \| 4.71 | 10 \| 2.43 | 9 \| 3.36 | 13 \| 5.14 | 12.0 \| 3.91 |
| **Human** | Expert-X | 199 | 19 | 29 | 134 | 95.25 |
| | Expert-Y | 152 | 47 | 17 | 500 | 179.0 |
| | Expert-Z | 96 | 13 | 32 | 500 | 160.3 |
| | Max \| Mean | 199 \| 149 | 47 \| 26.3 | 32 \| 26.0 | 500 \| 378 | 194.5 \| 144.8 |

Table 12: Same as Table 11, but for asynchronous affordance association.

inconsistent tendencies that they improve performance in different cases. For instance, Gemini-1.5-Flash with StuctM in paired fresh cooked attribute association leads other MLLM and other memory strategies, while QWen2-VL with ChainM in shake-cut action association makes leading. This further highlights the varying comprehension abilities of MLLMs when presented with the same input contexts, which also provides insight for improving the understanding of language at specific dimensions.

**Comparison of Different Categories.** Different paired categories exhibit specific abilities in asynchronous association. For instance, the paired categories of "fresh-cooked" and "painted-rusty" have the mean-step of asynchronous step on open-source MLLMs lower than 2.0, while the "fresh-cooked" and "painted-rusty" have the mean-step of asynchronous step greater than 3.0. We speculate that this is due to two different aspects reasons. One is the insufficient perception of specific categories within one concept, and the other is the lack of consciousness to convert the concept.

### E.4 DETAILED ABLATION STUDY

In this section, we supplement the experiment with the ablation study from the main text, concluding that the memory strategy with three examples shows the most comprehensive performance across all experiments. We below analyze the differences between different categories.

Table 14 shows the detailed results of the ablation study of example sizes. We observed that while the average max and mean step slightly differ between different example sizes, there are notable gaps within specific paired categories. For instance, in the case of LLaVA-OneVision with paired attribute "furry-metal", StructM-3 outperforms StructM-5 by approximately the mean-step of 1.95. Furthermore, while an average mean step of 3 outperforms in most cases, the example size of 1 and 5 also leads in certain cases. For instance, with QWen2-VL on "painted-rusty" affordance and LLaVA-OneVision on "fresh-cooked" attribute, the example size of 5 outperforms the others.

| Model | Memory | Asynchronous Verb Concept Association Chain (Max \| Mean Step) | | | | |
| --- | --- | --- | --- | --- | --- | --- |
| | | run-hit | drive-dress | cooking-build | shake-cut | Avg. |
| LLaVA-OneVision | NoM | 20 \| 2.74 | 21 \| 2.65 | 29 \| 5.44 | 20 \| 2.78 | 22.5 \| 3.40 |
| | StructM | 20 \| 2.39 | 14 \| 2.24 | 35 \| 4.16 | 21 \| 2.59 | 22.5 \| 2.85 |
| | NLM | 31 \| 2.87 | 35 \| 2.85 | 31 \| 4.96 | 15 \| 3.05 | 28.0 \| 3.43 |
| | ChainM | 21 \| 2.22 | 23 \| 2.39 | 23 \| 4.54 | 23 \| 2.87 | 22.5 \| 3.01 |
| QWen2-VL | NoM | 37 \| 4.65 | 19 \| 2.56 | 31 \| 5.99 | 23 \| 4.73 | 27.5 \| 4.48 |
| | StructM | 22 \| 2.51 | 23 \| 2.34 | 28 \| 4.52 | 23 \| 3.90 | 24.0 \| 3.32 |
| | NLM | **44** \| 4.97 | **36** \| 3.94 | 63 \| 8.31 | **35** \| 5.43 | **44.5** \| 5.66 |
| | ChainM | 28 \| 4.06 | 20 \| 3.34 | 44 \| 7.80 | 31 \| **5.79** | 30.8 \| 5.25 |
| mPLUG-Owl3 | NoM | 12 \| 1.17 | 12 \| 1.09 | 20 \| 2.20 | 16 \| 1.76 | 15.0 \| 1.56 |
| | StructM | 15 \| 1.75 | 15 \| 1.72 | 28 \| 4.84 | 21 \| 2.74 | 19.8 \| 2.76 |
| | NLM | 10 \| 1.46 | 16 \| 1.37 | 31 \| 4.17 | 20 \| 2.86 | 19.3 \| 2.47 |
| | ChainM | 16 \| 1.59 | 13 \| 1.43 | 28 \| 4.43 | 28 \| 2.95 | 21.3 \| 2.60 |
| **MoE** | NoM | 39 \| 4.13 | 23 \| 3.15 | 35 \| 6.39 | 20 \| 3.89 | 29.3 \| 4.39 |
| | StructM | 32 \| 2.74 | 21 \| 2.58 | 35 \| 4.75 | 22 \| 3.42 | 27.5 \| 3.37 |
| | NLM | 41 \| 4.40 | 29 \| 3.50 | 39 \| 6.55 | 30 \| 4.24 | 34.8 \| 4.67 |
| | ChainM | 31 \| 3.55 | 27 \| 3.19 | 40 \| 6.93 | 29 \| 4.29 | 31.8 \| 4.49 |
| **Gemini-1.5-Flash** | NoM | 27 \| 5.40 | 15 \| 3.90 | 15 \| 4.86 | 16 \| 3.03 | 18.3 \| 4.30 |
| | StructM | 16 \| 3.97 | 21 \| 3.47 | **75** \| 10.6 | 12 \| 4.63 | 31.0 \| **5.67** |
| | NLM | 13 \| 3.90 | 13 \| 2.70 | 39 \| **11.7** | 20 \| 4.23 | 18.0 \| 5.63 |
| | ChianM | 25 \| 5.50 | 14 \| 2.90 | 18 \| 5.07 | 17 \| 5.20 | 18.5 \| 4.67 |
| **GPT-4V** | NoM | 17 \| 4.45 | 7 \| 1.45 | 20 \| 4.60 | 14 \| 2.90 | 14.5 \| 3.35 |
| | StructM | 22 \| 4.10 | 17 \| **3.95** | 35 \| 6.80 | 14 \| 3.85 | 22.0 \| 4.68 |
| | NLM | 33 \| **6.70** | 18 \| 3.60 | 39 \| 8.11 | 13 \| 3.28 | 25.8 \| 5.42 |
| | ChainM | 16 \| 3.75 | 7 \| 2.50 | 26 \| 4.30 | 11 \| 3.40 | 15.0 \| 3.49 |
| **Human** | Expert-X | 64 | 12 | 39 | 12 | 30.5 |
| | Expert-Y | 10 | 32 | 26 | 59 | 30.5 |
| | Expert-Z | 19 | 17 | 77 | 35 | 39.5 |
| | Max \| Mean | 64 \| 31.0 | 32 \| 20.3 | 77 \| 47.3 | 59 \| 35.3 | 58.0 \| 33.5 |

Table 13: Same as Table 11, but for paired action synchronous association.

| Model | Memory | Paired Attribute Concept Association Chain (Max \| Mean Step) | | | | |
| --- | --- | --- | --- | --- | --- | --- |
| | | furry-metal | fresh-cooked | natural-ripe | painted-rusty | Avg. |
| LLaVA-OneVision | StructM-1 | 11 \| 1.28 | **36** \| 3.41 | **27** \| 1.18 | 47 \| 7.02 | 30.3 \| 3.22 |
| | StructM-3 | **41** \| **3.42** | 13 \| 2.41 | 22 \| **2.92** | 29 \| 3.82 | 26.3 \| 3.14 |
| | StructM-5 | 12 \| 1.47 | 34 \| 3.61 | 17 \| 1.36 | 67 \| 6.24 | 32.5 \| 3.17 |
| | NLM-1 | 14 \| 1.27 | 32 \| 4.08 | 11 \| 1.07 | 76 \| **9.26** | 33.3 \| 3.92 |
| | NLM-3 | 17 \| 1.36 | 32 \| 4.24 | 15 \| 1.19 | 83 \| 8.97 | **36.8** \| **3.94** |
| | NLM-5 | 16 \| 1.31 | 33 \| **4.27** | 11 \| 1.15 | **86** \| 8.86 | 36.5 \| 3.90 |
| | ChainM-1 | 12 \| 1.27 | 28 \| 3.30 | 12 \| 1.06 | 61 \| 7.73 | 28.3 \| 3.34 |
| | ChainM-3 | 11 \| 1.21 | 32 \| 3.47 | 17 \| 1.08 | 67 \| 7.65 | 31.8 \| 3.35 |
| | ChainM-5 | 11 \| 1.21 | 22 \| 3.18 | 15 \| 1.01 | 83 \| 7.99 | 32.8 \| 3.35 |
| Qwen2-VL | StructM-1 | 13 \| 1.26 | 21 \| 2.63 | 19 \| 1.19 | 33 \| 3.13 | 21.5 \| 2.05 |
| | StructM-3 | 12 \| 1.33 | 19 \| 2.55 | 18 \| 1.43 | 38 \| 3.75 | 21.8 \| 2.26 |
| | StructM-5 | 15 \| 1.36 | 19 \| 2.39 | 18 \| 1.37 | 34 \| 3.26 | 21.5 \| 2.09 |
| | NLM-1 | 15 \| 1.42 | **44** \| 4.59 | 16 \| 1.24 | 72 \| 10.18 | 36.8 \| 4.36 |
| | NLM-3 | 16 \| 1.63 | 43 \| **4.69** | 19 \| 1.49 | **97** \| 9.91 | **43.8** \| 4.43 |
| | NLM-5 | 13 \| 1.54 | 32 \| **4.69** | 16 \| 1.35 | 94 \| **10.52** | 38.8 \| **4.53** |
| | ChainM-1 | 11 \| 1.38 | 29 \| 4.19 | 11 \| 1.14 | 60 \| 6.94 | 27.8 \| 3.41 |
| | ChainM-3 | **25** \| **3.54** | 14 \| 2.79 | **20** \| **3.55** | 32 \| 3.93 | 22.8 \| 3.45 |
| | ChainM-5 | 11 \| 1.27 | 26 \| 4.51 | 17 \| 1.20 | 46 \| 6.58 | 25.0 \| 3.39 |

Table 14: Ablation study of the example sizes in synchronous and asynchronous association, which we compared by the results of Max | Mean Step. {Memory strategy}-X indicates the X example samples involved in the preview of the association. In this setting, we are implemented in open-source MLLMs LLaVA-OneVision and QWen2-VL with three memory strategies. The best and second results are shown in **bold** and underline, respectively.

| Model | Memory | Paired Action Concept Association Chain (Max \| Mean Step) | | | | |
|---|---|---|---|---|---|---|
| | | brush_hair-dive | clap-hug | shake_hands-sit | smoke-eat | Avg. |
| QWen-VL | NoM | 10 \| 1.02 | 11 \| 1.14 | 9 \| 1.04 | 10 \| 1.07 | 10.0 \| 1.07 |
| | StructM | 10 \| 1.03 | 11 \| 1.15 | 9 \| 1.17 | 10 \| 1.14 | 10.0 \| 1.12 |
| | NLM | 10 \| 1.04 | 11 \| 1.15 | 9 \| 1.11 | 10 \| 1.04 | 10.0 \| 1.09 |
| **Gemini-Pro-Vision** | NoM | 15 \| 6.00 | 10 \| 6.00 | 10 \| 4.20 | 5 \| 1.80 | 10.0 \| 4.50 |
| | StructM | **36** \| 10.5 | 24 \| 8.44 | 12 \| 4.20 | 14 \| 5.60 | 21.5 \| 7.19 |
| | NLM | 18 \| 8.10 | 20 \| 8.10 | **19** \| 5.70 | **31** \| **8.60** | 22.0 \| 6.25 |
| **GPT-4V** | NoM | 19 \| 6.20 | 5 \| 2.60 | 12 \| 4.30 | 9 \| 2.90 | 11.3 \| 4.00 |
| | StructM | 23 \| 12.0 | **37** \| **11.0** | 17 \| **7.70** | 14 \| 4.10 | **22.8** \| **8.70** |
| | NLM | 35 \| **15.6** | 17 \| 6.60 | 10 \| 3.10 | 19 \| 4.00 | 20.3 \| 7.18 |
| **Human** | Expert-X | 45 | 33 | 82 | 17 | 44.3 |
| | Expert-Y | 61 | 182 | 38 | 86 | 94.5 |
| | Expert-Z | 172 | 108 | 44 | 28 | 88.0 |
| | Max\| Mean | 172 \| 92.7 | 182 \| 107.7 | 82 \| 54.7 | 86 \| 43.7 | 130.5 \| 74.7 |

Table 15: The Max | Mean Step on paired action concept association from HMDB dataset in asynchronous association setting. We implemented at open-source MLLM QWen-VL, close-sourced MLLM Gemini-Pro-Vision, and GPT4-V, as well as human experts. The best and second results are shown in **bold** and underline, respectively.

## F  COMPARISON WITH MORE CONCEPTS

While we provide comprehensive experiments to benchmark the MLLM's performance in association tasks, we further expand an experiment on verb concepts to attain a solid perspective on association tasks. Specifically, we conduct asynchronous association on actions from the HMDB dataset, which is implemented in the open-source MLLM QWen-VL and closed-source MLLMs Gemini-Pro-Vision and GPT-4V. Simultaneously, we have integrated HMDB datasets into our human test interface. Furthermore, we include two memory strategies in this experiment, StructM, and NLM, as well as compare them against the baseline NoM.

Table 15 summarises the result of asynchronous association on the action from the HMDB dataset. Actions in the HMDB dataset are relatively easier than the actions in the Pangea dataset, which is reflected in the comparison of the mean-step at human experts between the different datasets in Table 13 and Table 15. Additionally, the open-source MLLM QWen-VL has a significant gap between closed-source MLLMs Gemini-Pro-Vision and GPT-4V. We speculate that this is due to the earlier version of QWen-VL, which had limited capabilities in multi-image perception. This has been effectively improved in the new QWen2-VL version, as demonstrated by the comparison between Table 13 and Table 15.

## G  COMPARISON WITH UNFILTERED RAW DATA

| Model | MLLM Verification | Individual Attribute Concept Association Chain (Max \| Mean Step) | | | | |
|---|---|---|---|---|---|---|
| | | NoM | StructM | NLM | ChainM | Avg. |
| LLaVA-OneVision | No | 26.8 \| 3.05 | 27.1 \| 3.36 | 25.9 \| 3.05 | 29.1 \| 3.10 | 27.2 \| 3.14 |
| | Yes | 48.0 \| 5.30 | 60.6 \| 5.80 | 75.4 \| 8.91 | 60.8 \| 7.03 | 61.2 \| 6.76 |

Table 16: Comparison of whether MLLM verification on the individual attribute association. We are implemented in the LLaVA-OneVision with three memory strategies and one baseline. The comparison of "Yes" and "No" indicates the effectiveness of the MLLM verification in data refinement.

It is noted that we involve three comprehensive steps in data refinement to ensure the data quality is met in the association task. We below involve the experiment for comparison of the effectiveness and difference with and without the MLLM Verification step.

Table 16 and 17 summarise the comparison of results for individual concept or paired concepts association, respectively. For comparison of individual concept association, we utilize LLaVA-OneVision with four memory strategies. Additionally, for paired concepts association, we provide a broader implementation that includes two open-source MLLMs, LLaVA-OneVision and QWen-VL,

| Model | Concept | MLLM Verification | Paired Concept Association Chain (Max \| Mean Step) | | | | |
|---|---|---|---|---|---|---|---|
| | | | NoM | StructM | NLM | ChainM | Avg. |
| LLaVA-OneVision | Attribute | No | 20.0 \| 2.12 | 22.3 \| 2.47 | 25.3 \| 2.78 | 25.8 \| 2.43 | 23.3 \| 2.45 |
| | | Yes | 23.0 \| 2.65 | 30.5 \| 3.13 | 36.8 \| 3.94 | 31.8 \| 3.35 | 30.5 \| 3.27 |
| | Affordance | No | 13.5 \| 1.91 | 16.0 \| 2.09 | 11.8 \| 2.06 | 12.5 \| 2.05 | 13.5 \| 2.03 |
| | | Yes | 23.3 \| 3.11 | 26.3 \| 3.14 | 21.5 \| 3.59 | 21.3 \| 3.29 | 23.1 \| 3.28 |
| | Action | No | 18.0 \| 2.60 | 18.8 \| 2.73 | 20.5 \| 2.67 | 17.3 \| 2.70 | 18.7 \| 2.68 |
| | | Yes | 22.5 \| 3.40 | 22.5 \| 2.85 | 28.0 \| 3.43 | 22.5 \| 3.01 | 23.9 \| 3.17 |
| QWen2-VL | Attribute | No | 22.3 \| 2.47 | 18.0 \| 2.01 | 31.8 \| 3.22 | 23.0 \| 2.76 | 23.8 \| 2.62 |
| | | Yes | 25.5 \| 3.28 | 21.8 \| 2.26 | 43.8 \| 4.43 | 28.5 \| 3.45 | 29.9 \| 3.35 |
| | Affordance | No | 16.5 \| 2.13 | 15.3 \| 1.87 | 23.3 \| 2.60 | 17.0 \| 2.63 | 18.0 \| 2.31 |
| | | Yes | 20.8 \| 3.13 | 20.0 \| 2.28 | 23.3 \| 3.63 | 22.8 \| 3.45 | 21.7 \| 3.12 |
| | Action | No | 21.5 \| 3.19 | 19.8 \| 2.74 | 27.8 \| 3.76 | 23.8 \| 3.67 | 23.2 \| 3.34 |
| | | Yes | 27.5 \| 4.48 | 24.0 \| 3.32 | 44.5 \| 5.66 | 30.8 \| 5.25 | 31.7 \| 4.68 |

Table 17: Same as Table 16, but for comparison on the Paired categories attribute association. Notably, we expand the model to LLaVA-OneVision and QWen2-VL with three different concepts.

with three different concepts. In this section, we mainly compare the results of with or without MLLM Verification, *i.e.*, comparing the line of "Yes" to the line of "No".

From the result, we easily find that the result with MLLM verification outperforms without MLLM verification, *i.e.*, the line of "Yes" larger than "No". This demonstrates that MLLM Verification effectively reduces the sample with potentially confusing annotations or those requiring powerful advanced perception abilities beyond the current capabilities of MLLMs. More interesting, while the MLLM Verification step reduces the complexity of perception on object concept understanding, we also observe a similar pattern between with and without MLLM verification, *i.e.*, the tendency is consistent when comparing each line.

# H    ETHIC REVIEW

As shown in Figure 8, in our human evaluation protocol, we have implemented a comprehensive Ethic Report mechanism to proactively address and mitigate potential ethical concerns. The user interface incorporates dedicated options for participants to report issues related to privacy or other ethical considerations. This is facilitated through a structured reporting system comprising categorical buttons and an open-ended text field for additional context. This approach enables active participant engagement in ethical oversight during the evaluation phase, fostering a collaborative approach to responsible AI development. We prioritize transparency by empowering participants to articulate concerns about potential privacy infringements, algorithmic bias, or instances where the system may induce discomfort or exhibit opaque behavior. This methodology aligns with the best practices delineated by Zaldivar et al. (2019) Kennedy-Mayo & Gord (2024), which emphasizes the criticality of integrating transparency and user feedback mechanisms to ensure fairness and accountability in machine learning systems.

Besides the ethical review implemented within our human testing demo, it is crucial to emphasize that our approach builds upon the previous ethical review of the original datasets. They have undergone a rigorous ethical review process, particularly concerning data sourcing, privacy considerations, and bias mitigation. This prior evaluation also sets a strong foundation for ethical safeguards.

