# OpenReview forum: "The Labyrinth of Links: Navigating the Associative Maze of Multi-modal LLMs"
_ICLR.cc/2025/Conference — ICLR 2025 Poster_

### Official Review · Reviewer_hJvy · 2024-10-17

**Soundness:** 3
**Presentation:** 3
**Contribution:** 3
**Rating:** 8
**Confidence:** 3

**Summary:**

The paper proposes a benchmark, using an annotation-free construction method to transform general datasets. The benchmark includes three levels of association tasks: single-step, synchronous, and asynchronous associations. The authors conduct extensive experiments involving multiple open-source and closed-source MLLMs, including state-of-the-art models like GPT-4V, and compare their performance with human experts.

**Strengths:**

The paper offers a fresh approach to assessing MLLMs by focusing on their associative abilities, which is a novel contribution to the field. The annotation-free construction method for association tasks is innovative and has the potential to simplify the creation of benchmarks in this area.

**Weaknesses:**

The evaluation is limited to MLLMs’ zero-shot ability in association tasks across adjectives and verb semantic concepts. This may not fully capture the complexity of real-world scenarios where MLLMs are expected to perform.

While the paper analyzes failure cases in the association process, it could have provided a more in-depth understanding of why these failures occur.

The paper focuses on single-step, synchronous, and asynchronous associations, which are complex tasks. However, it might not fully capture the nuances of human associative learning, which often involves more gradual and contextually influenced processes.

**Questions:**

How might the authors' findings on the associative capabilities of MLLMs influence the design of future models, particularly in terms of memory and reasoning architectures?

Are there any specific areas of application where the authors believe the current gaps in associative capabilities are particularly problematic, and thus, warrant immediate attention in research?

How did you ensure that the limitations of the original datasets did not significantly affect the results?

---

> ### Author Response · Authors · 2024-11-24
>
> ***Question 1:*** Why was the evaluation limited to MLLMs’ zero-shot ability in association tasks across adjectives and verb semantic concepts?
>
> ***Answer:***
>
> - The association is using prior memory, knowledge, or experience to build the link between different things,  which is practically built on similar characteristics or features that contribute to its identity or functionality. These features are formally defined as action, attribute, and affordance in an object, which are adjectives or verbs at the semantic level. Hence, we developed the benchmark based on adjectives and verbs in our paper. Apart from adjectives and verbs, noun association also builds on adjectives and verbs. Adverb lacks individual expression, which is under-explored in the community.
> - We think association will inspire the community to explore a new learning paradigm that learning breaks through the dependence on match-based or interleaved-based data using any unpaired data association learning. This will thoroughly solve the data hunger bottleneck of the community, and push the MLLM close to human intelligence. Hence, we concentrate on defining the association and investigating the deficiency of MLLMs in this paper.
>
> ***Question 2:*** There needs in-depth analysis of why failures occur.
>
> ***Answer:*** We speculate three possible reasons leading to the failure of MLLMs. We also expand **Figure 10** in the supplementary to clarify this analysis.
> - **The association needs more precise locationality.** Compared to existing methods[1] that rely on language questions with explicit information to retrieve related image content, our benchmark includes a comprehensive memory that stores structured knowledge from prior experiences. The key information relevant to the model’s predictions is often distributed across different locations and represented as relatively local, subtle elements within the context. This highlights the need for more advanced attention in our benchmark.
> - **The association needs more common sense-based reasoning.**  The association is an implicit link between the different objects that are not explicitly shown in the query, requiring a deeper common sense-based reasoning. For instance, existing benchmarks[2] infer the weather in the image, they directly use the word in the query to retrieve the part in the image. In contrast, our benchmark only specifies the direction of the link, requiring the model to think of the underlying link within the prompted scope.
> - **The association needs short-term memory, which is underexplored by existing work.** Our association benchmark is sequential concept links between different objects, which inherently need the memory mechanism to retain the prior experiences. This aspect of memory remains underexplored in existing benchmarks, with current designs falling significantly behind the sophistication of human cognitive systems. This inspires the future study in memory design.
>
>
> [1]. MMRA: A benchmark for evaluating multi-granularity and multi-image relational association capabilities in large visual language models.
> [2]. Milebench: Benchmarking mllms in long context.
>
> ***Question 3:*** The paper focuses on single-step, synchronous, and asynchronous associations, which are complex tasks. However, it might not fully capture the nuances of human associative learning, which often involves more gradual and contextually influenced processes.
>
> ***Answer:*** Thanks for your valuable suggestion for further improving our benchmark.
> - The association implemented in our benchmark is the simpler version that makes decisions based on prior experience and current observation. This design eliminates the influence of complex, gradual, and contextually influenced processes,  focusing only on the static concept.
> - Incorporating gradual and context-driven processes would enhance the complexity of the tasks, demanding advanced precise locationality and memory mechanisms. Simultaneously, this progress requires re-designing data that requires high cost to collect, which is beyond the scope of our paper.
> - We will sincerely consider and discuss your suggestion, and develop a more strict benchmark for more advanced MLLMs in future work.

---

> > ### Author Response · Authors · 2024-11-24
> >
> > ***Question 4*** How might the authors' findings on the associative capabilities of MLLMs influence the design of future models, particularly in terms of memory and reasoning architectures?
> >
> > ***Answer:*** For future MLLM, we think there exist two potential solutions.
> > - **Learning beyond paired data.** The human association capability indicates that humans can learn by any missing modality data and associate it with prior knowledge. It is intuitively suggested that next-generation MLLM should not be dependent on matched-based or interleaved-based data rather than any unpaired data that may have missing modality. We sincerely believe that our paper will provide the deficiency of current MLLM, which will pave the way for next-generation MLLM.
> > - **Short-term memory rather than long-context.** Memory is a pivotal module in the human recognition system, which transfers any experience into structured information. This process involves mapping overlapped information to existing information and expanding new information into appropriate locations. That is also crucial for next-generation MLLM, which links all information with prior experience all the time. This improvement will break through the dependence on match-based or interleaved-based data, solving the data hunger bottleneck in the whole community.
> >
> > ***Question 5:*** Are there any specific areas of application where the authors believe the current gaps in associative capabilities are particularly problematic, and thus, warrant immediate attention in research?
> >
> > ***Answer:***
> > - In robotics, there exist two main challenges: the scarcity of data and the limited generalization across diverse environments, embodiments, and tasks. More specifically, human demonstration data for imitation learning is negligible compared to vision-language scenarios. The learned model is only effective in the same setting that is reflected in execute will failure when the weather of the environment and location of robots changes.
> > - We think that the association will push the utilization of any unpaired data, solving data hunger. Furthermore, strengthening associations across environments and embodiments will enhance the model’s generalization capabilities.
> >
> > ***Question 6:*** How did you ensure that the limitations of the original datasets did not significantly affect the results?
> >
> > ***Answer:***
> > - We propose an annotation-free data construction method that can easily transfer raw supervised data from association tasks. To further control the quality and remove any bias in the raw dataset, we developed three strict steps to eliminate potential bias for the association, including an image resolution filter, MLLM verification, and human expert evaluation. For enhanced clarity, we have included **Figure 7** in the supplementary.
> > - More specifically,  the image resolution filter eliminates low-resolution images. Next, MLLM verification employs advanced Gemini and GPT models to confirm the presence of ground-truth labels within the images, filtering out those requiring higher perceptual capabilities. Finally, 100 testers interact with our benchmark filter to identify and report any biased or ethically sensitive images.

---

> > > ### Comment · Area_Chair_5mqY · 2024-11-26
> > >
> > > Dear Reviewer hJvy, the ICLR discussion period is extended. Could you please take a look at the authors' rebuttal and other reviews, and see whether you would like to update your ratings? The authors would greatly appreciate your consideration and responses.

---

> > > > ### Comment · Reviewer_hJvy · 2024-11-30
> > > > **Official Comment by Reviewer hJvy**
> > > >
> > > > Thank you for your rebuttal. I have no further questions. I will increase my score.

---

> > > > > ### Author Response · Authors · 2024-11-30
> > > > >
> > > > > Thank you again for the time and effort you have devoted to providing thoughtful reviews.

---

### Official Review · Reviewer_6DyH · 2024-10-31

**Soundness:** 2
**Presentation:** 2
**Contribution:** 3
**Rating:** 5
**Confidence:** 5

**Summary:**

This paper proposes a novel benchmark aimed at evaluating the association capabilities of Multi-modal Large Language Models (MLLMs), a crucial yet often overlooked aspect of human intelligence. The authors formulate a specific association task based on adjective and verb semantic concepts and introduce an innovative annotation-free method for constructing these tasks, minimizing the reliance on expensive data annotation. They implement a rigorous data refinement process to enhance dataset clarity and present three levels of association tasks: single-step, synchronous, and asynchronous. The investigation covers a wide range of MLLMs, including both open-source and closed-source models, exploring various memory strategies and involving human experts. Findings reveal a significant performance gap between current MLLMs, including state-of-the-art models like GPT-4V, and human capabilities in association tasks. It implies that this benchmark could advance future MLLM research.

**Strengths:**

1. **Originality**:
   - The benchmark specifically targeting association capabilities represents a unique contribution to the MLLM literature. While benchmarking is prevalent, focusing on association adds a novel dimension to the evaluation of language models.

2. **Quality**:
   - The authors' annotation-free construction method for association tasks is a practical innovation that alleviates the common challenges associated with extensive manual data labeling, enhancing the quality and usability of the benchmark.

3. **Clarity**:
   - The paper is well-structured and clearly articulated, with straightforward definitions and explanations of association tasks. This clarity facilitates comprehension and accessibility for a broad audience.

4. **Significance**:
   - By highlighting the substantial gap between MLLM performance and human intelligence in association tasks, the paper underscores the importance of developing models that can better mimic human cognitive capabilities. This sets the stage for future research efforts in enhancing MLLMs.

**Weaknesses:**

1. **Limited Novelty**:
   - The benchmarking of association capabilities is not entirely novel. Similar efforts can be found in previous research focusing on common sense reasoning, such as "CommonSenseQA" and "HellaSwag," which also evaluate reasoning and associative capabilities.

2. **Subjectivity in Tasks**:
   - Association tasks can be influenced by subjective interpretations, yet the paper does not sufficiently address how such subjectivity is mitigated. Discussion around the prompt design for MLLMs—especially concerning analogy tasks—could enhance the validity of the results.

3. **Resource Intensiveness**:
   - The comprehensive nature of the study, involving multiple models and memory strategies, raises concerns regarding reproducibility. The resource demands may hinder wider adoption among researchers with limited computational resources.

4. **Experimental Design**:
   - The rationale for selecting specific MLLMs could be more thoroughly explained, and comparisons with existing benchmarks would strengthen the justification for the new benchmark's necessity.

5. **Discussion Depth**:
   - The discussion section could delve deeper into the implications of the findings, particularly regarding practical applications and theoretical contributions to AI and cognitive modeling.

6. **Performance Gap Exploration**:
   - While the performance gap between MLLMs and human intelligence is significant, the paper could provide more in-depth analysis of the causes and potential paths toward bridging this gap.

7. **Bias in Dataset Annotation**:
   - The paper lacks clarity on how biases in the initial dataset annotations are addressed, which could affect the robustness of the findings.

**Questions:**

1. **Rationale for Focus**: What is the rationale behind selecting adjectives and verbs as the focal point for your semantic concepts in the association tasks?

2. **Bias Mitigation**: Regarding equation (1), where \(z_{ij}\) is constrained to 0 or 1, could this introduce bias from human evaluations? How can this bias be effectively removed from the assessment?

3. **Overlooked Capability**: Can you elaborate on why you believe the association ability of MLLMs has been overlooked in previous research?

4. **Novelty Clarification**: How does your proposed benchmark differentiate itself from existing benchmarks assessing association and reasoning capabilities in language models?

5. **Bridging Performance Gaps**: Given the performance gap observed, what advancements do you foresee as crucial for future MLLMs to approach human-level performance in association tasks?

6. **Resource Considerations**: How can researchers with limited resources effectively utilize your benchmark, considering its resource-intensive nature?

7. **Model Selection Criteria**: What criteria informed your selection of specific MLLMs for this study, and how do you view these models in the context of the current state of MLLM research?

8. **Future Implications**: How do you envision your findings influencing the design of future MLLMs and the benchmarks used for their evaluation? What next steps do you recommend in this line of research?

9. **Evolving MLLM Performance**: As MLLMs continue to evolve and improve in association tasks, how do you anticipate the relevance of your findings will change?

---

> ### Author Response · Authors · 2024-11-24
>
> ***Question 1 (Rationale for Focus):*** What is the rationale behind selecting adjectives and verbs as the focal point for your semantic concepts in the association tasks?
>
> ***Answer:***
> - The association is using prior memory, knowledge, or experience to build the link between different things,  which is practically built on similar characteristics or features that contribute to its identity or functionality. These features are formally defined as action, attribute, and affordance in an object, which are adjectives or verbs at the semantic level.
> - Apart from adjectives and verbs, noun association also builds on adjectives and verbs. Adverb lacks individual expression, which is under-explored in the community.
>
> ***Question 2 (Bias Mitigation):***  Regarding equation (1), where (z_{ij}) is constrained to 0 or 1, could this introduce bias from human evaluations? How can this bias be effectively removed from the assessment?
>
> ***Answer:***
> - We propose an annotation-free data construction method that can easily transfer raw supervised data for association tasks. To further control the quality and remove any bias in the raw dataset, we developed three strict steps to eliminate potential bias for the association, including an image resolution filter, MLLM verification, and human expert evaluation. For enhanced clarity, we have included **Figure 7** in the supplementary.
> - More specifically, the image resolution filter eliminates low-resolution images. Next, MLLM verification employs advanced Gemini and GPT models to confirm the presence of ground-truth labels within the images, filtering out those requiring higher perceptual capabilities. Finally, 100 testers interact with our benchmark filter to identify and report any biased or ethically sensitive images.
>
>
> ***Question 3 (Overlooked Capability and Novelty Clarification):*** Can you elaborate on why you believe the association ability of MLLMs has been overlooked in previous research? The benchmarking of association capability is not entirely novel. Similiar efforts can be found in previous research. How does your proposed benchmark differentiate itself from existing benchmarks assessing association and reasoning capabilities in language models?
>
> ***Answer:*** Thanks for your advice about further clarifying the difference with existing work. We have added a comparison **Figure 9** in the revised paper's supplementary material.
>
> - Our paper is the first to conduct a comprehensive investigation into sequential concept association in MLLMs.
> - As summarized in a recent benchmark survey[1], MLLMs' benchmark mainly concentrates on the design of more complex tasks and evaluating nuanced correlations between input samples.
> - More deep comparison, our benchmark also needs general perception capability as LLaVA-Bench[2], nuanced features within images as Compbench[4], and cooperation across different modalities as SpatialRGPT[5]. Furthermore, our benchmark needs exceptional abilities beyond existing work. First, our benchmark builds on adjectives and verbs that need deeper perception than nouns in MMVP[3]. Second, our benchmark is sequential images beyond the two images in the existing work of MILEBENCH[6]. Finally, our benchmark breaks the closed reasoning into open scenarios, which use practice to verify model ability instead of direct answering in existing work as MARVEL[7].
> - CommonsenseQA and HellaSwag concatenate on commonsense inference in input samples in language and multi-modal settings, respectively. This capability is also involved in our benchmark and we consider exceptional abilities.
> - In summary, our benchmark involves the basic capability existing in the previous benchmark, including general perception, nuance features, and cooperation across different modalities. Simultaneously, our benchmark involves exceptional abilities, including deep concept perception, sequential image tasks, and larger solution space.
>
> [1]. A Survey on Benchmarks of Multimodal Large Language Models.
>
> [2]. Visual Instruction Tuning
>
> [3]. Eyes Wide Shut? Exploring the Visual Shortcomings of Multimodal LLMs
>
> [4]. Compbench: A comparative reasoning benchmark for multimodal llms
>
> [5]. SpatialRGPT: Grounded Spatial Reasoning in Vision-Language Models
>
> [6]. MILEBENCH: Benchmarking MLLMs in Long Context
>
> [7]. MARVEL: Multidimensional Abstraction and Reasoning through Visual Evaluation and Learning

---

> > ### Author Response · Authors · 2024-11-24
> >
> > ***Question 4 (Bridging Performance Gaps & Future Implications):*** Given the performance gap observed, what advancements do you foresee as crucial for future MLLMs to approach human-level performance in association tasks? How do you envision your findings influencing the design of future MLLMs and the benchmarks used for their evaluation? What next steps do you recommend in this line of research?
> >
> > ***Answer:***
> > - We speculate three possible reasons contributing to the significant gap between MLLMs and humans. We also expand **Figure 10** in the supplmentary to clarify this analysis.
> >     - **The association needs more precise locationality.** Compared to existing methods[8] that rely on language questions with explicit information to retrieve related image content, our benchmark includes a comprehensive memory that stores structured knowledge from prior experiences. The key information relevant to the model’s predictions is often distributed across different locations and represented as relatively local, subtle elements within the context. This highlights the need for more advanced attention in our benchmark.
> >     - **The association needs more common sense-based reasoning.**  The association is an implicit link between the different objects that are not explicitly shown in the query, requiring a deeper common sense-based reasoning. For instance, existing benchmarks[9] infer the weather in the image, they directly use the word in the query to retrieve the part in the image. In contrast, our benchmark only specifies the direction of the link, requiring the model to think of the underlying link within the prompted scope.
> >     - **The association needs short-term memory, which is underexplored by existing work.** Our association benchmark is sequential concept links between different objects, which inherently need the memory mechanism to retain the prior experiences. This aspect of memory remains underexplored in existing benchmarks, with current designs falling significantly behind the sophistication of human cognitive systems. This inspires the future study in memory design.
> > - For future MLLM, we think there exist two potential solutions.
> >     - **Learning beyond paired data.** The human association capability indicates that humans can learn by any missing modality data and associate it with prior knowledge. It is intuitively suggested that next-generation MLLM should not be dependent on matched-based or interleaved-based data rather than any unpaired data that may have missing modality. We sincerely believe that our paper will provide the deficiency of current MLLM, which will pave the way for next-generation MLLM.
> >     - **Short-term memory rather than long-context.** Memory is a pivotal module in the human recognition system, which transfers any experience into structured information. This process involves mapping overlapped information to existing information and expanding new information into appropriate locations. That is also crucial for next-generation MLLM, which links all information with prior experience all the time. This improvement will break through the dependence on match-based or interleaved-based data, solving the data hunger bottleneck in the whole community.
> >
> > [8]. MMRA: A benchmark for evaluating multi-granularity and multi-image relational association capabilities in large visual language models.
> > [9]. Milebench: Benchmarking mllms in long context.
> >
> > ***Question 5 (Resource Considerations):*** How can researchers with limited resources effectively utilize your benchmark, considering its resource-intensive nature?
> >
> > ***Answer:***
> > - Our benchmark is built with a convenient annotation-free construction method that does not require high-cost human labor re-annotation. Additionally, our benchmark can easily generalized to more complex association tasks without any extra cost.
> > - In addition, our benchmark will inspire the study of learning in unpaired data, which means using missing modality with association ability to improve the model's capability. Any new learning paradigm will originate some small evidence, this provides a giant potential within limited resources.

---

> > > ### Author Response · Authors · 2024-11-24
> > >
> > > ***Question 6 (Model selection criteria)*** What criteria informed your selection of specific MLLMs for this study, and how do you view these models in the context of the current state of MLLM research?
> > >
> > > ***Answer:***
> > > - Our benchmark evaluates multiple open-source and closed-source MLLM. Closed-source MLLM includes advanced Gemini and GPT4-V. Open-source MLLM selection with the following considerations.
> > > - With the advent of the most advanced MLLM this year, the new open-sourced MLLM explores the capability of multi-image or video input. Simtaneously, our benchmark mainly evaluates the MLLM with multiple image inputs.
> > > - Building on these foundations, we employ a prioritized approach to select open-source MLLMs based on three progressively diminishing criteria:
> > >     - The model’s capability to handle multi-image or video inputs.
> > >     - Performance on general MLLM leaderboards, such as [OpenCompass Leaderboard](https://rank.opencompass.org.cn/leaderboard-multimodal/?m=REALTIME).
> > >     - The model’s structural design, including variations in LLM architectures and training paradigms, such as from-scratch training versus post-training adaptations of pre-existing LLMs
> > > - Based on these strategies, we select LLaVA-OneVision, QWen2-VL, and mPLUG-Owl3 as our open-source MLLMs.
> > >
> > > ***Question 7 (Evolving MLLM Performance):*** As MLLMs continue to evolve and improve in association tasks, how do you anticipate the relevance of your findings will change?
> > >
> > > ***Answer:***
> > > - With the development of advanced MLLM, we anticipate the model will gradually improve capability in common sense understanding and locationality. These improvements also improve the model's performance in association tasks, but it will not attain human intelligence for the following reasons.
> > >     - The improvement will be reflected in two key aspects: incorporating additional modalities and enhancing reasoning capabilities. Our benchmark encompasses multi-dimensional reasoning, where improvements in input reasoning alone are insufficient to resolve the questions effectively.
> > >     - In addition,  the efficient learning ability of human intelligence is the association of new knowledge with prior experience, which also enables humans to learn from any data. Hence, using any unpaired data to develop next-generation MLLM is important.

---

> > > > ### Comment · Area_Chair_5mqY · 2024-11-26
> > > >
> > > > Dear Reviewer 6DyH, the ICLR discussion period is extended. Could you please take a look at the authors' rebuttal and other reviews, and see whether you would like to update your ratings? The authors would greatly appreciate your consideration and responses.

---

> > > > > ### Comment · Reviewer_6DyH · 2024-11-26
> > > > > **Comment to authors**
> > > > >
> > > > > I appreciate your detailed and honest response and would like to comment.
> > > > >
> > > > > **Question 1 (Rationale for Focus)**:
> > > > > Focusing on adjectives and verbs is well-justified, as they are key for expressing object attributes (adjectives) and actions (verbs), essential for defining object identity and functionality in association tasks. Although adverbs are underexplored, they could provide valuable insights into context and temporal relationships in multimodal reasoning, making them an area worth investigating further.
> > > > >
> > > > > **Question 2 (Bias Mitigation)**:
> > > > > Your multi-step bias mitigation process, including image resolution filters, MLLM verification, and expert evaluation, is solid. The use of Gemini and GPT for image validation is an excellent approach, though it would be interesting to further explore how these models handle subtle cultural or ethical biases. Including human evaluators adds crucial nuance to the process.
> > > > >
> > > > > **Question 3 (Overlooked Capability and Novelty Clarification)**:
> > > > > Your focus on sequential concept association and the incorporation of adjectives and verbs for deeper semantic reasoning sets your benchmark apart. Unlike existing benchmarks that prioritize noun-based associations, your approach explores the complex relationships between objects across time, enhancing multimodal reasoning. The benchmark's ability to handle sequential images and open-ended reasoning further differentiates it from previous work.
> > > > >
> > > > > **Question 4 (Bridging Performance Gaps & Future Implications)**:
> > > > > You rightly identify gaps in locational precision, common sense reasoning, and short-term memory. Moving beyond paired data and enhancing memory mechanisms are key to bridging the gap between current models and human-like association capabilities. Your suggestions for improving memory and reasoning reflect critical areas for future MLLM development, particularly in enabling models to handle real-world, unpaired data.
> > > > >
> > > > > **Question 5 (Resource Considerations)**:
> > > > > The annotation-free construction method is a smart solution for reducing resource needs, while still enabling complex association tasks. Your emphasis on learning from unpaired data offers exciting potential for more efficient model training, making it accessible to researchers with limited resources.
> > > > >
> > > > > **Question 6 (Model Selection Criteria)**:
> > > > > Your criteria for selecting MLLMs—focusing on multi-image/video input capability, leaderboard performance, and architectural variety—are well-founded. This ensures a comprehensive evaluation across both open- and closed-source models, which is crucial for advancing multimodal reasoning and association tasks.
> > > > >
> > > > > **Question 7 (Evolving MLLM Performance)**:
> > > > > As MLLMs evolve, improvements in common sense reasoning and multimodal integration will enhance association tasks. However, achieving human-level performance will require models that not only process richer data but also incorporate flexible, context-sensitive learning. The shift toward unpaired data and memory mechanisms will be key to overcoming the current limitations and bridging the gap with human cognition.

---

> ### Author Response · Authors · 2024-11-26
>
> Dear Reviewer 6DyH,
>
> We sincerely appreciate you recognizing some highlights in our paper, we will thoroughly your suggestions and polish our paper in the camera-ready version. In addition, if you have any concerns or additional suggestions, please share them with us at any time. We will sincerely consider your suggestion and conduct possible experiments to address your concerns.
>
> Thank you again for your time and effort.
>
> Best regards,
>
> Authors

---

### Official Review · Reviewer_8My5 · 2024-11-04

**Soundness:** 3
**Presentation:** 1
**Contribution:** 2
**Rating:** 6
**Confidence:** 3

**Summary:**

This paper proposed an evaluation benchmark, which evaluates multimodal large language models' performance on predicting association in three scenarios: single-step association, synchronized multi-step associations, and asynchronized multi-step associations.

**Strengths:**

1. This paper proposed a novel task and a perspective on MLLMs.

2. Various MLLMs are tested on the benchmark.

**Weaknesses:**

1. The definition of the task "deduction" in Table 1 could be better explained and illustrated like the "association" task in Figure. 2.

2. It would be essential to include human performance in Table 1 to understand current gap between MLLMs and human. In addition, the authors might consider including more recent and powerful GPT-4o performance in Table 1.

3. For Table 1, since MLLMs' performance on the association task is rather high (around 80\% accuracy), there could be concerns about the potential improvement MLLMs can achieve for future works. To understand such, human performance might be a valuable baseline to refer to.

4. The evaluation settings in the synchronous association task in Figure 3 could be difficult to understand. The authors might consider better explaining their inputs and expected outputs and their metric calculation as some pseudo-code.

5. The performance comparison in Figure 3 is not very insightful, which might lacks proper explanations about why MLLMs are significantly inferior to human judgement. Such significant gap could raise concerns about the effectiveness of the proposed evaluation protocol.

6. The illustration of Figure 6 is not very self-explainable, while its textual explanations in context are also very brief, which could hinder readers understanding of the major concerns in current MLLMs on such tasks.

**Questions:**

1. Could the authors explain the input and expected output of the single-step "deduction" task? Since the benchmark is claimed to evaluate MLLMs' ability in association prediction, whether deduction is part of the benchmark? In addition, could the authors explain the relation between the "association" and "deduction" tasks?

2. For the results in Figure 3, since the human expert could achieve hundreds rounds of association, could the authors explain how the length of association is evaluated? Specifically, is the benchmark contains the ground truth of hundreds rounds of associations? Is each prediction on the next association necessarily requires all the previous memories? Are those MLLMs prompted with such long-term association or with only current inputs?

3. What would be the prompting format for such tasks. Are those images concatenated as a single image or separately input into the prompt? Are those images interleaved with proper textual instructions, so that MLLMs could understand the intention correctly? In addition, is the proposed tasks formatted as multi-choice question-answering tasks, where generated answers could be well mapped to ground-truth?

---

> ### Author Response · Authors · 2024-11-24
>
> ***Question 1:*** The definition of 'deduction' in Table 1. And, the relation of 'association' step and 'deduction' step. In addition, the input and expected output of the single-step 'deduction' step.
>
> ***Answer:*** Thanks for your suggestion, we will further revise the content to eliminate potential confusion. We add some content and figures below to solve your concerns.
> - The **association** is using prior memory, knowledge, or experience to build the link/relationship between different things, which can be realized in different modalities and domains, abstract concepts with concrete things, etc. The **association benchmark** proposed in this paper is the **sequential concept link** between current observation and prior experiences, which intuitively includes making predictions with association memory (prior experience) and deduction potential experience in this process.
> - Hence, the 'deduction step' in our benchmark is using the "guessed" underlying link from the association to predict the next concept.
> - We also add **Figure 9** in the supplementary for comparing with existing work and input and expected output of our benchmark. In the 'deduction step', the input is the query image,  the correctly selected image in the association step, and an instruction to guide the model to generate the appropriate link between two images.
>
> ***Question 2:*** There lack of human performance in Table 1. The performance of the association task is rather high in Table 1, there exists some concern for future improvement.
>
> ***Answer:***
> - In our experiment, we also include users to evaluate the human performance in the single-step association setting, which shows the corrected link and input question. The test results indicate that only 1-2 users made fewer than 5 errors across 100 rounds, while all other users completed the tasks flawlessly. Therefore, in this paper, we have taken **human performance in single-step association to be 100%**.
> - As shown in an existing benchmark[1][2], with the advanced models developed, the improvement of performance is gradually slow. The findings suggest that achieving scores from 0 to 80 is relatively straightforward, whereas advancing from 80 to 100 is significantly more challenging. This pattern underscores distinct tendencies across different score ranges.
> - Our benchmark develops multiple tasks that increase the complexity across simple to complex.  Single-step is the most simple task in our benchmark where humans are unlikely to make errors.  With the gradual enhancement of MLLM, we can concentrate on the complex task that also proposes some difficulties for human experts. Additionally, the proposed benchmark can infinitely improve the complexity, i.e., involves more concepts in association links.
>
> [1]. MILEBENCH: Benchmarking MLLMs in Long Context
>
> [2]. MMIU: Multimodal Multi-image Understanding for Evaluating Large Vision-Language Models
>
> ***Question 3:*** The evaluation setting is hard to understand in Figure 3. It is recommend to expand some pseudo-code for clarity. Is the benchmark contains the ground truth of hundreds rounds of associations? Is each prediction on the next association necessarily requires all the previous memories? Are those MLLMs prompted with such long-term association or with only current inputs?
>
> ***Answer:***
> - Transferring from the single-step to synchronous and asynchronous steps, we evaluate the association capability in a dynamic sequential environment. This involves iteratively processing the association step and deduction step and exiting when the association makes an error. In this setting, we evaluate the max / mean step, which is the maximum step and average step across multiple round tests, respectively.
> - To save memory during synchronous and asynchronous steps, we developed three distinct strategies for transferring experiences in the deduction step to update association memory. These strategies encompass memory storage, deep memory retention, and forgetting mechanisms.
> - We have provided expanded pseudo-code for the task in Subsection A.3 of the supplementary.

---

> > ### Author Response · Authors · 2024-11-24
> >
> > ***Question 4:*** In Figure 3, there lacks proper explanations about why MLLMs are significantly inferior to human judgetment.
> >
> > ***Answer:*** We speculate three possible reasons contributing to the significant gap between MLLMs and humans. We also expand **Figure 10** to clarify this analysis.
> > - **The association needs more precise locationality.** Compared to existing methods[3] that rely on language questions with explicit information to retrieve related image content, our benchmark includes a comprehensive memory that stores structured knowledge from prior experiences. The key information relevant to the model’s predictions is often distributed across different locations and represented as relatively local, subtle elements within the context. This highlights the need for more advanced attention in our benchmark.
> > - **The association needs more common sense-based reasoning.**  The association is an implicit link between the different objects that are not explicitly shown in the query, requiring a deeper common sense-based reasoning. For instance, existing benchmarks[4] infer the weather in the image, they directly use the word in the query to retrieve the part in the image. In contrast, our benchmark only specifies the direction of the link, requiring the model to think of the underlying link within the prompted scope.
> > - **The association needs short-term memory, which is underexplored by existing work.** Our association benchmark is sequential concept links between different objects, which inherently need the memory mechanism to retain the prior experiences. This aspect of memory remains underexplored in existing benchmarks, with current designs falling significantly behind the sophistication of human cognitive systems. This inspires the future study in memory design.
> >
> > [3]. MMRA: A benchmark for evaluating multi-granularity and multi-image relational association capabilities in large visual language models.
> > [4]. Milebench: Benchmarking mllms in long context
> >
> > ***Question 5:*** The illustration of Figure 6 is not very self-explanable, while its textual explanation in context are also very brief.
> >
> > ***Answer:*** Thanks for your valuable suggestion, we have already revised the figure.
> > - Figure 6 shows that attention weights under two different memory strategies. They exhibit the consistent phenomenon across two memory strategies that MLLMs tend to assign higher attention weights to tokens closer to the last position, a pattern that deviates significantly from human behavior. This shows the inconsistency that important areas should get higher attention, i.e., the frame in the figure should be dark. These findings further support the speculation that the deficiency of MLLMs is limited attention.
> >
> >
> > ***Question 6:*** What is the prompt format in this task? The format of input image? Additionally, whether the tasks formatted as multi-choice question-answering tasks? And, whether the generated answers could be well mapped to ground-truth?
> >
> > ***Answer:***
> > - In the experiment, we evaluate association capability with question-answer type. The input of MLLMs is taken popular interleaved format. In addition, we devise output instructions to guide the output of MLLMs based on powerful MLLMs' instruction-following ability.
> > - More specifically, we have taken synchronous association with structured memory as an example. The input is Given the memory: {eat: [sandwich, pizza]} \n Determine the relationship between the original image and the candidate images, and select the images with the same attribute as the original image. Original image:<image>. Candidate images: Image1: <image>, Image2: <image>.\n Your response should be direct and exclusively only include one of the following items.\n Options: [Image1, Image2]. The more detailed prompt format for all settings can be found in Table 3 in the supplementary.
> >
> > ***Question 7:*** The author might consider including more recent and powerful GPT-4o performance in Table 1.
> >
> > ***Answer:*** The development of MLLMs is advancing at an exceptionally rapid pace.  We have made our best efforts to include the latest models. We have also added GPT-4o in our single-step association setting.
> > - The table below shows the expanded result in advanced GPT-4o, which evaluates the single-step success ratio within each type of concept.
> >
> >
> > | GPT4-o     | Association |         |       |        | Deduction |
> > | ---------- | ----------- | ------- | ----- | ------ | --------- |
> > | Concept    | NoM         | StructM | NLM   | ChainM | -         |
> > | Attribute  | 84.49       | 88.83   | 89.87 | 78.86  | 78.14     |
> > | Affordance | 84.93       | 85.36   | 86.76 | 81.54  | 27.30     |
> > | Action     | 86.97       | 88.72   | 86.13 | 85.90  | 57.28     |

---

> > > ### Comment · Reviewer_8My5 · 2024-11-25
> > >
> > > Thanks for the authors' comprehensive responses. I believe this work is solid with comprehensive empirical results. However, the presentation of this work is very difficult to understand, especially regarding the highly intrinsic settings. In addition, while the authors provided several potential reasons contributing to the significant gap between MLLMs and humans, the real reasons behind such phenomenon are still mystical. For example, could the interleaved prompt format be one of the major factors, since interleaved MLLM reasoning ability cannot be easily acquired? Would some simple agentic framework be able to solve such tasks by engineering designs and/or heuristic algorithms? Some first-attempt solutions might shed some light and give some hope for future work. I raised my score to 6 due to the authors' hard work in providing comprehensive results.

---

> ### Author Response · Authors · 2024-11-26
>
> Dear Reviewer 8My5,
>
> We sincerely appreciate the reviewer's engagement and valuable comments, which are instrumental in enhancing the quality of our work. We will enhance the presentation of our paper and polish related figures in the camera-ready version. Besides, we have finished the results of ablating the interleaved prompt format. We will continue to investigate any potential aspects for enhancing our work.
>
> **Implementation:** Our prompt comprised four parts, including task instruction, memory, input observation, and output instruction. It is noted that if output instruction is not in the last position, the MLLM will lose instruction-following capability and output the non-predefined format of 'Image 1 ' or 'Image 2'. Hence, we ablate other potential orders: memory_first (existing result): memory, task instruction, input observation, and output instruction.  instruction_first: task instruction, memory, input observation, and output instruction. memory_observation_first: memory, input observation, task instruction, and output instruction.
>
> | LLaVA-OneVision | prompt_type           | StructM    | NLM       | ChainM    | Avg.      |
> | --------------- | --------------------- | ---------- | --------- | --------- | --------- |
> |                 | memory_first          | 60.6/5.80  | 75.4/8.91 | 60.8/7.39 | 65.6/7.37 |
> |                 | instruction_first     | 47.0/5.48  | 53.9/6.29 | 40.3/5.15 | 47.1/5.64 |
> |                 | memory_question_first | 48.85/5.97 | 76.0/9.01 | 55.5/6.92 | 60.1/7.3  |
>
> **Conclusion:** From the above result, we can easily conclude that the significant gap between MLLMs and humans is unrelated to the interleaved prompt format. More specifically, instruction_first has a lower performance than others. The prompt used in our paper (memory_first) has a close performance with the memory_question_first type.
>
> Excitingly, the Author-Reviewer discussion phase has been extended. If you have any additional concerns or exceptional suggestions, please share them with us at any time. We will sincerely consider your concerns and conduct potential experiments to enhance our work.
>
> Best regards,
>
> Authors
>
> **Detailed Results:**
>
>
> | LLaVA-OneVision | memory type | prompt type           | furry   | metal   | fresh   | cooked  | natural | ripe    | painted  | rusty     | Avg.       |
> | --------------- | ----------- | --------------------- | ------- | ------- | ------- | ------- | ------- | ------- | -------- | --------- | ---------- |
> |                 | StructM     | memory_first          | 15/1.38 | 30/4.03 | 41/4.75 | 32/4.04 | 13/1.21 | 51/6.51 | 65/9.23  | 238/15.24 | 60.6/5.80  |
> |                 |             | memory_question_first | 15/1.23 | 28/3.66 | 46/5.17 | 38/4.25 | 10/1.14 | 57/8.05 | 53/7.91  | 129/12.43 | 47.0/5.48  |
> |                 |             | instruction_first     | 15/1.38 | 43/4.07 | 51/4.86 | 39/3.93 | 18/1.14 | 76/7.07 | 53/8.95  | 96/16.34  | 48.85/5.97 |
> |                 | NLM         | memory_first          | 14/1.46 | 40/5.24 | 68/6.09 | 48/3.93 | 12/1.16 | 91/8.88 | 81/10.64 | 249/33.88 | 75.4/8.91  |
> |                 |             | memory_question_first | 13/1.30 | 35/4.35 | 50/5.63 | 31/3.89 | 10/1.08 | 73/8.32 | 86/9.15  | 133/16.64 | 53.9/6.29  |
> |                 |             | instruction_first     | 15/1.43 | 48/5.34 | 67/6.12 | 44/4.20 | 13/1.21 | 83/8.79 | 81/10.05 | 257/34.95 | 76.0/9.01  |
> |                 | ChainM      | memory_first          | 16/1.49 | 63/4.66 | 58/5.23 | 28/3.29 | 11/1.13 | 63/6.89 | 70/9.70  | 177/23.79 | 60.8/7.39  |
> |                 |             | memory_question_first | 11/1.29 | 25/3.62 | 37/4.95 | 29/3.50 | 11/1.08 | 56/6.79 | 49/6.99  | 105/13.04 | 40.3/5.15  |
> |                 |             | instruction_first     | 14/1.42 | 37/4.32 | 41/5.35 | 34/3.48 | 17/1.12 | 65/6.35 | 80/9.82  | 156/23.54 | 55.5/6.92  |

---

### Official Review · Reviewer_Lt2g · 2024-11-07

**Soundness:** 3
**Presentation:** 2
**Contribution:** 3
**Rating:** 8
**Confidence:** 5

**Summary:**

This work proposes a new benchmark testing the zero-shot association ability of MLLMs. Association origins from object concept learning, where the task is to connect observations with previous practice memory by identifying the underlying principle. For example, images of fresh apples,  oranges, vegetables could be connected through the adjective "fresh". It is a fundamental capability for humans. The proposed benchmark leverage previous datasets of object concept learning and is created in an annotation-free way. Basically, the labels in datasets of object concept learning directly provide the underlying principle (concept) that could connect objects.

The authors designed different settings to test MLLMs' zero-shot association ability: single-step vs multi-step, synchronous vs asynchronous. According to the reported results, all the leading MLLMs show a gap in terms of the association ability, compared to humans.

**Strengths:**

**originality**
The originality is good. The work proposed and will open-source a new benchmark for MLLMs, by transforming a previous ML benchmark into a LLM benchmark.

**significance**
It shows all MLLMs have an obvious gap vs humans. In this sense, the benchmark is able to evaluate and push an overlooked capability towards AGI.

**Weaknesses:**

**quality**
There are two aspects that improvements could be made on. First, regarding error analysis, more insights are preferred. For example, by "limited perception capability", is it due to the limitation of image encoder/resolution or something intrinsic to LLMs. Most public MLLMs are composed of a separate image encoder+adaptor and the main LLMs. Some ablation studies on this aspect are preferred. Second, checking the correlation between this new benchmark and existing benchmarks is preferred. For example, if the performance on this new benchmark is strongly correlated with a weighted sum of those on some existing benchmarks, we could better know how to improve such a capability. If no correlation, this work might point out an overlooked dimension, which could also motivate more related benchmarks to be created.

**clarity**
The presentation could be improved a bit, by adding more clear examples. For example, Figure 7 in the context is better in explaining the exact task than Figure 1. And adding figures for the annotation-free transformation from object concept learning datasets can better explain the exact dataset building process.

**Questions:**

1. I don't see a clear argument on why sticking to zero-shot setting? As there is a practice memory involved, is it also naturally similar to few-shot?
2. Why does the study only focus on MLLMs? Is it because the object concept learning dataset is for multi-modal initially? How hard is it to make a text-only corresponding benchmark?

---

> ### Author Response · Authors · 2024-11-24
>
> ***Question 1***: Ablation study for limited perception capability.
>
> ***Answer:*** Thanks for your valuable suggestion. Below, we add related experiments to investigate MLLM's perception further.
>
> - **Implementation:** To deeply analyze the perception capabilities of MLLM, we assess the representation capability by isolating specific components of the architecture. Specifically, we test the performance of the image encoder and the image projector in isolation, where they generate image and text representations. These representations are then compared using cosine similarity. In contrast, we assess task performance when using whole parameters, i.e., using MLLM as a question-answer agent. The experiments are conducted on the OCL dataset, where accuracy is computed for object category understanding. For attributes and affordances, we calculate mean Average Precision (mAP) to evaluate multi-class performance.
> - **Result analysis**: It is important to note that it is feasible to rigorously compare the performance at the representation and task level, which is only taken as a reference. From the results below, we can easily find that the result of the image encoder and image projector is lower than MLLM in the object category, which reflects that **MLLM has advanced semantic understanding ability compared to only the image encoder and realigned representation with image projector**. However, for attribute and affordance understanding, all methods exhibit relatively weak performance. It is evident that current **MLLMs have limited perception capabilities in the properties of an object**, prompting the need to focus on selected concepts for association.
>
>
> | Concept          | Image Encoder (Contrastive) | Image Projector (Contrastive) | MLLM (QA) |
> | ---------------- | --------------------------- | ----------------------------- | --------- |
> | Category (Acc)   | 12.93                       | 0.79                          | 81.96     |
> | Attribute (mAP)  | 9.81                        | 7.91                          | 10.52     |
> | Affordance (mAP) | 25.30                       | 24.15                         | 24.49     |
>
> ***Question 2***: Comprehensively compare with existing benchmark.
>
> ***Answer:*** Thanks for your advice about further clarifying the difference with existing work. We have added a comparison **Figure 9** in the revised paper's supplementary material.
>
> ***Answer:***
> - Our paper is the first to conduct a comprehensive investigation into sequential concept association in MLLMs.
> - As summarized in a recent benchmark survey[1], MLLMs' benchmark mainly concentrates on the design of more complex tasks and evaluating nuanced correlations between input samples.
> - More deep comparison, our benchmark also needs general perception capability as LLaVA-Bench[2], nuanced features within images as Compbench[4], and cooperation across different modalities as SpatialRGPT[5]. Furthermore, our benchmark needs exceptional abilities beyond existing work. First, our benchmark builds on adjectives and verbs that need deeper perception than nouns in MMVP[3]. Second, our benchmark is sequential images beyond the two images in the existing work of MILEBENCH[6]. Finally, our benchmark breaks the closed reasoning into open scenarios, which use practice to verify model ability instead of direct answering in existing work as MARVEL[7].
> - In summary, our benchmark involves the basic capability existing in the previous benchmark, including general perception, nuance features, and cooperation across different modalities. Simultaneously, our benchmark involves exceptional abilities, including deep concept perception, sequential image tasks, and larger solution space.
>
> [1]. A Survey on Benchmarks of Multimodal Large Language Models.
>
> [2]. Visual Instruction Tuning
>
> [3]. Eyes Wide Shut? Exploring the Visual Shortcomings of Multimodal LLMs
>
> [4]. Compbench: A comparative reasoning benchmark for multimodal llms
>
> [5]. SpatialRGPT: Grounded Spatial Reasoning in Vision-Language Models
>
> [6]. MILEBENCH: Benchmarking MLLMs in Long Context
>
> [7]. MARVEL: Multidimensional Abstraction and Reasoning through Visual Evaluation and Learning

---

> > ### Author Response · Authors · 2024-11-24
> >
> > ***Question 3***: More figures for the annotation-free transformation.
> >
> > ***Answer***: Thanks for your advice, we append **Figure 7** in the main paper.
> >
> >
> > ***Question 4***: Why is the paper sticking to a zero-shot setting? As there is a practice memory involved, is it naturally similar to the few-shot?
> >
> > ***Answer:***
> > - Thank you for your valuable suggestion. Our benchmark is non-totally consistent with the few-shot setting.  The difference is that our benchmark retains non-directly related experience rather than a consistent information structure and template shared with subsequent questions.
> > - More detailedly,  our benchmark is to find potential links/relations between different objects there without a ground-truth answer. Memory is the experience in inductive and deductive, instead of direct information structure and template for subsequent questions.
> > - Our benchmark firstly formally defines the association question, there lack of strict division for this setting. We will carefully consider your advice and further refine the expression of our setting.
> >
> > ***Question 5***:
> > Why does the study only focus on MLLMs? Is it because the object concept learning dataset is for multi-modal initially? How hard is it to make a text-only corresponding benchmark?
> >
> > ***Answer:*** Thank you propose a crucial question for the broad significance of our benchmark. We have added some statements and experiments hope to further discuss with you.
> >
> > - **Association is essentially multi-modal.** The association is using prior memory, knowledge, or experience to build the link between different things, which can be realized in different modalities, different domains, abstract concepts with concrete things, etc. Hence, the association is not limited to one specific modality rather than the plentiful world with multiple modalities.  Building on this insight, we create a multi-modal association benchmark for current advanced MLLM.
> > - **Unreason between association and language-only setting.** Our benchmark can be seamlessly extended to LLM using the widely adopted method used in the community that utilizes the advanced image caption model BLIP-2[8] to transfer images to textual descriptions.  However, there will exist some uncertainty when associated only with language. More specifically,
> >     - The concept expressed via language is usually more abstract and precise than images. This highlights that images provide more imagination than language, which is the foundation for association.
> >     - The association within language is close to synonym search, which is more likely the correlation between contexts rather than association.
> > - **Exploratory experiments and analysis.** Regardless of potential risk in language-only association, we process object images through the BLIP-2 model to generate captions. We first analyze the overlap ratio between generation captions with ground truth labels. We then access language-only association on the LLaMA3.1-8B-Instruct model.
> >     - **Generated data analysis.** The overlap ratio is a measure used to see if a caption correctly mentions the ground-truth concept. The result below reflects the statistical overlap between the generated caption and ground truth label. The findings indicate that image captioning performs appropriately decent in generating correct categories but exhibits significantly poor performance in capturing attributes and affordances. This underscores the limitations of the advanced image captioning model in these aspects.
> >     - **Experiment result analysis.** Building on the generated LLM association benchmark, we evaluate the LLM association capability on LLaMA3.1-8B-Instruct, which is the leading model in the open-source LLM leaderboard, in the synchronous attribute concept association setting.  From the below result, we can easily find the association in language-only settings has no effect, i.e., memory strategies lower or close to no memory. This supports the speculation that association realized in language-only setting is correlation between input context.
> >
> > | Generated Data analysis | category | attribute | affordance |
> > | ----------------------- | -------- | --------- | ---------- |
> > |                         | 60.39    | 0.71      | 0.21       |
> >
> > | Experiment Result | Model                | Modality   | NoM         | StructM   | NLM         | ChainM    |
> > | ----------------- | -------------------- | ---------- | ----------- | --------- | ----------- | --------- |
> > | Existing Result   | LLaVA-OneVision      | Multimodal | 48.0/5.3    | 60.6/5.80 | 75.4/8.91   | 60.8/7.03 |
> > |                   | QWen2-VL             |            | 85.5 / 10.0 | 37.9/3.57 | 122.5/13.96 | 56.9/7.39 |
> > |                   | mPLUG-Owl3           |            | 13.4/1.27   | 34.4/3.69 | 34.0/3.42   | 31.5/3.37 |
> > | New Result        | LLaMA3.1-8B-Instruct | Language   | 16.3/1.68   | 7.4/0.47  | 9.38/0.83   | 15.8/1.31 |

---

> > > ### Author Response · Authors · 2024-11-30
> > >
> > > Dear Reviewer Lt2g,
> > >
> > > We sincerely appreciate the time and effort you have devoted to providing thoughtful reviews and valuable feedback. As the Author-Reviewer discussion phase is nearing its conclusion, if you have any concerns or additional suggestions, please share them with us at any time. We will sincerely consider your suggestion and conduct possible experiments to address your concerns.
> > >
> > > Thank you again for your time and effort.
> > >
> > > Best regards,
> > >
> > > Authors

---

### Author Response · Authors · 2024-11-24

Dear Reviewers,

We sincerely appreciate the time and effort you have devoted to providing thoughtful reviews and valuable feedback.
We carefully analyzed each weakness and question, addressing them thoroughly. We also refined and added some figures in the revised version of the paper. The main concerns are addressed in the following ways:
- Detailed data curation and quality control (Figure 7).
- Comprehensive comparison with the existing benchmark (Figure 9).
- Deep analysis of MLLM inferior humans (Figure 10).

We hope these revisions and discussions have adequately addressed your concerns. Expecting your further discussion or suggestion to enhance our work.

Best regards,

Authors

---

### Meta-Review · Area_Chair_5mqY · 2024-12-20

**Metareview:**

This submission introduces a benchmark to test the zero-shot association abilities of multimodal LLMs. The authors argue that the association task, which involves linking observations to prior practice memory, is a fundamental human capability, e.g., connecting images of fresh apples, oranges, and vegetables through the adjective "fresh." By leveraging existing datasets from object concept learning, the proposed benchmark uses the labels in these datasets to provide the underlying principles connecting objects. The authors also designed different task settings to assess multimodal LLMs' performance on association: single-step vs. multi-step and synchronous vs. asynchronous tasks. The benchmarking finds existing LLMs show significant gaps in association abilities compared to humans.

The strengths of this work includes:
1) it introduces a new benchmark for evaluating the association abilities of multi-modal LLMs;
2) The benchmark discovers the significant gap between multimodal LLMs and human in associative tasks;
3) The authors also introduce an annotation-free method for dataset construction, making it relatively easier and cost-effective to create benchmarks for evaluating association tasks.

Weaknesses:
1) it lacks detailed error analysis, particularly on the limitations of image encoders or LLMs. It could benefit from ablation studies to identify the causes of failures.
2) it doesn't sufficiently explore how the new benchmark correlates with existing benchmarks or provide deeper analysis of the performance gap between multimodal LLMs and humans.
3) its presentation could be improved, as pointed out by the reviewers: some figures are not clear enough and could benefit from more detailed explanations and illustrations to improve understanding.

Overall, the final ratings this work received are towards positive: 8, 8, 6, 5. The authors' rebuttal managed to convince the reviewers and make reviewers  Lt2g and hJvy increase the ratings. Reviewer 6DyH (with rating 5) gave a detailed response to authors' rebuttal and confirmed that many issues / questions raised were addressed (however, no rating update is given). All factors considered, this work is recommended for an acceptance.

**Additional Comments On Reviewer Discussion:**

The authors' rebuttal managed to convince the reviewers and make reviewers  Lt2g and hJvy increase the ratings. Though no rating update is given), reviewer 6DyH (with rating 5) gave a detailed response to authors' rebuttal and confirmed that many issues / questions raised were addressed.

The authors' rebuttal is detailed and associated with many new experimental results. The reviewers' responses are straightforward and brief --- increasing the ratings. The discussion is rather not particularly enthusiastic.

---

### Decision · Program_Chairs · 2025-01-22

Accept (Poster)